# Rho-dependent transcription termination proceeds via three routes

Eunho Song 1, Heesoo Uhm 1,4, Palinda Ruvan Munasingha 2, Seungha Hwang 3, Yeon-Soo Seo[2], Jin Young Kang 3✉, Changwon Kang 2✉ & Sungchul Hohng 1✉

Rho is a general transcription termination factor in bacteria, but many aspects of its mechanism of action are unclear. Diverse models have been proposed for the initial interaction between the RNA polymerase (RNAP) and Rho (catch-up and stand-by pre-terminational models); for the terminational release of the RNA transcript (RNA shearing, RNAP hyper-translocation or displacing, and allosteric models); and for the post-terminational outcome (whether the RNAP dissociates or remains bound to the DNA). Here, we use single-molecule fluorescence assays to study those three steps in transcription termination mediated by *E. coli* Rho. We find that different mechanisms previously proposed for each step co-exist, but apparently occur on various timescales and tend to lead to specific outcomes. Our results indicate that three kinetically distinct routes take place: (1) the catch-up mode leads first to RNA shearing for RNAP recycling on DNA, and (2) later to RNAP displacement for decomposition of the transcriptional complex; (3) the last termination usually follows the stand-by mode with displacing for decomposing. This three-route model would help reconcile current controversies on the mechanisms.

[1] Department of Physics and Astronomy, and Institute of Applied Physics, Seoul National University, Seoul 08826, Republic of Korea. [2] Department of Biological Sciences, Korea Advanced Institute of Science and Technology, Daejeon 34141, Republic of Korea. [3] Department of Chemistry, Korea Advanced Institute of Science and Technology, Daejeon 34141, Republic of Korea. [4] Present address: Department of Physics, University of Oxford, Oxford OX1 3PU, UK. ✉email: jykang59@kaist.ac.kr; ckang@kaist.ac.kr; shohng@snu.ac.kr

Transcription termination is essential for precise expression and proper regulation of genes. In bacteria, intrinsic termination requires only *cis*-acting elements on nascent RNAs, while factor-dependent termination relies on both *cis*-acting RNA elements and a *trans*-acting termination factor, Rho (ρ). Bacterial ρ, participating in transcription termination[1] and many others[2,3], is an ATP-dependent RNA translocase[4]. ρ binds nascent RNA at *rut* (Rho utilization) sites using its primary RNA-binding surface in an open state and wraps the RNA with its secondary RNA-binding surface to transform into the active state[4].

With respect to the pre-terminational modes regarding how ρ encounters RNA polymerase (RNAP) or elongation complex (EC) in order to mediate termination, widely accepted is the catch-up mechanism, in which ρ first binds nascent RNA at a *rut* site and catches up with EC that is pausing at a termination site according to the so-called kinetic-coupling[5], RNA-dependent[6], RNA-centric[7], and tracking[8] models.

This classical mechanism for the pre-terminational step is challenged by the stand-by mechanism, in which ρ first binds RNAP of stable EC and stands by for binding an incipient RNA *rut* site emerging out of RNAP according to the so-called RNAP-dependent[6], EC-centric[7], and allosteric[9] models. This alternative pre-terminational mechanism has been supported by several biochemical[9–11] and structural data[7,12], although questioned by other biochemical data[13].

For the RNA release, which defines termination of transcription, the catch-up mode generally assumes a critical role in ρ's motor activity. The mechanical force exerted by ρ is expected to shear RNA off from RNA·DNA hybrid in the RNA shearing model or displace RNAP forward out of the ribonucleotide (NTP) incorporation site in the RNAP hyper-translocation model[14]. The stand-by mode, on the other hand, generally assumes that RNA is released or EC is disassembled not by the motor action of ρ but by allosteric changes in EC conformation triggered by ρ[7,9,12].

As to the post-terminational outcomes, recent single-molecule studies revealed that RNAPs mostly remain on DNA after RNA release at ρ-independent, intrinsic termination and are directly reused for reinitiation of transcription on the same DNA molecule[15–17]. After releasing RNA at termination, RNAPs diffuse on DNA in upstream and downstream directions, change the moving direction, and even flip themselves on DNA, until reinitiating transcription on promoters or falling off DNA[15–17]. This post-terminational one-dimensional (1D) diffusion of RNA-free RNAP on DNA is named 1D recycling, shortened here as recycling, and the reinitiation by such 1D recycling RNAP is named 1D reinitiation.

Much less frequently at intrinsic termination, EC decomposes into the three essential components all at once with simultaneous or near-simultaneous dissociations of RNA and RNAP from DNA. This one-step disassembly of EC is called here decomposing for short. While recycling is much more frequent than decomposing from intrinsic termination[15–17], it is unknown yet whether recycling, decomposing, or both result from ρ-dependent termination.

In this study using single-molecule fluorescence measurements, we discover that ρ-dependent termination of a single terminator follows three different routes rather than a single path while characterizing the termination events of individual transcription complexes in three steps; pre-terminational setup, terminational release, and post-terminational outcome. The three routes each lead to a specific destination on a distinct timescale.

## Results

### Single-molecule monitoring of transcription termination. We constructed a DNA template containing the T7A1 promoter and the *Salmonella mgtA* leader with a ρ-dependent terminator[18,19] (Fig. 1a, Supplementary Table 1). Its upstream end is labeled with biotin for immobilization and its downstream end with fluorescent Cy5. To generate stalled transcription complexes, we incubate a DNA template with fluorescent Cy3-5′-labeled ApU dinucleotide, unlabeled ATP, CTP, and *Escherichia coli* RNAP with σ70 in a transcription stalling buffer. Transcription initiates preferentially with Cy3-ApU but stalls with a short RNA (5′-Cy3-AUACC-3′) due to missing GTP, so active transcription complexes are labeled with two fluorescent dyes, Cy3 on RNA and Cy5 on DNA (Fig. 1a).

The fluorescent initially stalled complexes (ISCs) are immobilized on polymer-coated quartz slides using biotin–streptavidin conjugation and extensively washed to remove all unimmobilized complexes. In single-molecule termination assays (Fig. 1b), the immobilized ISCs are elongated by the addition of NTPs and ρ (100 nM unless varying), while the fluorescence of Cy3-RNA and Cy5-DNA in individual transcription complexes is monitored in real-time using total-internal-reflection fluorescence microscopy.

Vanishing of Cy3 spots after NTP + ρ injection indicates release of transcript RNA out of immobilized complexes resulting from transcription termination at the termination-associated pausing site (termination site) or from runoff transcription at the DNA downstream end following transcription readthrough at the termination site (Supplementary Fig. 1). This Cy3 vanishing is distinguished from photobleaching, which takes much longer than termination or runoff (Supplementary Fig. 2) and little affects the measurements of termination and runoff.

### Co-occurring fast and slow ρ-dependent terminations. Another noticeable fluorescence change was protein-induced fluorescence enhancement (PIFE). PIFE occurs to cyanine (Cy) dyes when their local environment becomes more viscous by binding proteins to hinder the dye's photoisomerization from fluorescent *trans*-isoform to non-fluorescent *cis*-isoform[20–22], as it has been seen with many DNA-binding and motor proteins[23,24].

In this study, Cy3 PIFE occurs when the Cy3 labeled at the 5′ end of RNA remains proximal to RNAP in ISCs, and the PIFE decreases when the 5′ end moves away from RNAP as the RNA grows in length upon resumption of transcription elongation from ISCs. Individual complexes showing this pattern of Cy3 fluorescence changes have gone through active transcription and therefore are solely included in our data analysis.

By contrast, Cy5 PIFE arises when the Cy5 labeled at the downstream end of DNA is approached by RNAP. Either transcribing RNAP with RNA after readthrough at the termination site or recycling RNAP without RNA after termination at the termination site can reach the downstream end to cause the PIFE. On the other hand, once RNAP falls off DNA, it readily diffuses away and seldom rebinds DNA even at the ends in this single-molecule assay.

Real-time monitoring of Cy3-RNA and Cy5-DNA signals revealed three distinct patterns of their fluorescence changes (Fig. 1c, Supplementary Fig. 3). The first pattern arises from the readthrough followed by runoff from DNA, as Cy5 PIFE starts before the Cy3 signal vanishes (Fig. 1c left, Supplementary Fig. 3a). Transcribing RNAP ignores the termination signal, passes beyond the termination site, reaches the DNA downstream end to bring about Cy5 PIFE, and finally dissociates from the DNA end, releasing a Cy3-labeled readthrough-product runoff-transcript RNA to make the Cy3 signal vanish.

The second pattern represents the termination resulting in RNAP recycling on DNA (called recycling termination), as Cy5 PIFE starts as soon as Cy3 signal vanishes (Fig. 1c center, Supplementary Fig. 3b). Only RNA is released at the termination

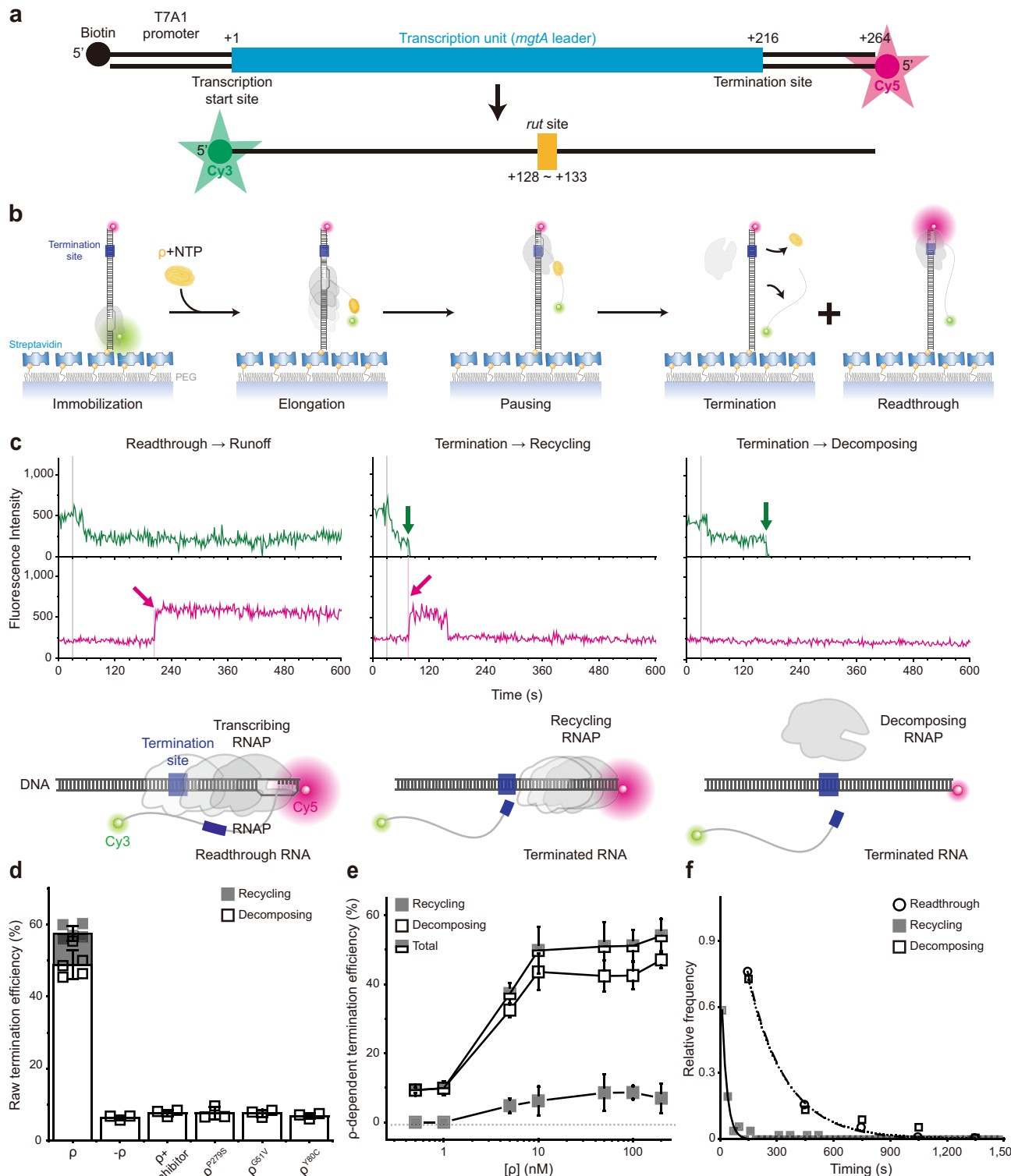

site, vanishing the Cy3 signal, and then post-terminational recycling RNAP diffuses on DNA to reach the downstream end and sticks there for a while, generating Cy5 PIFE.

The third pattern reflects the termination resulting in EC decomposing (called decomposing termination), as Cy3 signal disappears but Cy5 PIFE never arises (Fig. 1c right, Supplementary Fig. 3c). RNAP releases terminated RNA to vanish the Cy3 signal but does not reach the DNA downstream end to produce Cy5 PIFE. In this event, the three essential components of the transcription complex fall apart together for decomposing.

Alternatively, RNA-free RNAP only briefly remains on DNA upon termination and falls off DNA to diffuse away before reaching the downstream end.

The 1D reinitiation could occur by recycling RNAP after termination and subsequent backward diffusion to the promoter on the same DNA molecule[15–17]. It can be followed by the second-round transcription for termination or readthrough and subsequent such events, but these cyclic events are not monitored in this assay as the reinitiated transcripts are not labeled with Cy3 after the ISC-washing step.

**Fig. 1 Single-molecule assay of ρ-dependent termination. a** Transcription of a fluorescent Cy5-labeled DNA template with a ρ-dependent *mgtA* terminator produces RNA transcripts containing a *rut* site and labeled with fluorescent Cy3 at the 5′ end. **b** Scheme of single-molecule holistic assay of transcription termination. The occurrence of Cy3 and Cy5 PIFEs are indicated by enlarged green and magenta circles, respectively. **c** Three distinct patterns of fluorescence changes in active transcription. Readthrough (left, $n = 251$ complexes), recycling termination (center, $n = 51$), and decomposing termination (right, $n = 289$) are distinguished by their characteristic patterns in fluorescence time traces of Cy3 (green) and Cy5 (red) at Cy3 excitation (top) and Cy5 excitation (bottom). NTP + ρ were injected at 30 s (gray lines) after fluorescence monitoring starts. A schematic diagram is shown below each representative trace, and additional details are described in Supplementary Fig. 3. **d** Termination efficiencies (TEs) and background termination. Raw TEs are estimated as the ratio of decomposing or recycling events to the sum of all termination and readthrough events. Negative controls were performed without ρ or with ρ plus ρ-inhibitor bicyclomycin, and three ρ mutants exhibit only background levels of termination activity. **e** ρ-dependency of termination. The ρ-dependent TEs of recycling (solid) and decomposing (open) along with their sums (total) in the y-axis are plotted against input ρ concentrations on a base-10 log scale in the x-axis. ρ-dependent TEs are calculated by subtraction of the ρ-independent background TE (6.3%) from raw TEs. **f** Relative frequencies of readthrough, recycling termination, and decomposing termination timings. Termination was timed as the delay from Cy3 PIFE diminishing to Cy3 vanishing, whereas readthrough was timed as the delay from Cy3 PIFE diminishing to Cy5 PIFE starting. These timings were estimated with the data fitting to single exponential functions. Error bar represents the standard deviation of the mean from $n \geq 3$ independent experiments. The numbers of analyzed molecules for **d** and **e** are in Supplementary Table 2. Source Data file includes the data for (**d**–**f**).

This basic setup of single-molecule termination assay is called holistic assay to be distinguished from the two below-described assays that each measure only a certain discrete mode of termination. In a negative control experiment without ρ, termination efficiency (TE) drops to a background level, $6.3 \pm 0.7\%$ (Fig. 1d). This ρ-independent TE is subtracted from the raw TEs to yield ρ-dependent TEs in all the following estimations. Additionally, with ρ plus ρ-inhibitor bicyclomycin, TE drops to a similar background level (Fig. 1d).

In each assay, every transcription complex is counted as readthrough, recycling termination, or decomposing termination. Their relative frequencies are $42.6 \pm 3.4\%$, $8.6 \pm 2.0\%$, and $48.8 \pm 3.9\%$, respectively as measured in three independent experiments with 10 mM $Mg^{2+}$. Thus, raw TE is $57.4 \pm 3.4\%$ (sum of the recycling and decomposing terminations, Fig. 1d) or ρ-dependent TE is 51%, which is comparable to a previous measurement[18] at 3.5 mM $Mg^{2+}$.

Interestingly, decomposing outcome dominates over recycling outcome from ρ-dependent termination and only decomposing is observed in background terminations (Fig. 1d), in contrast to intrinsic termination, where recycling outcome is predominant[15–17]. In titration experiments, decomposing and recycling TEs both steeply increase on rising ρ concentration (Fig. 1e) demonstrating the ρ dependence, but the ratio of decomposing and recycling TEs little changes (Supplementary Fig. 4) implying that the two are both intrinsic outcomes of ρ-dependent termination.

Interestingly, recycling termination occurs much earlier (27 s) than decomposing termination (199 s) or readthrough (198 s) as shown in Fig. 1f. Termination was timed as the interval from the Cy3 PIFE diminishing timepoint to the Cy3 signal vanishing timepoint, whereas readthrough was timed as the delay from Cy3 PIFE diminishing to Cy5 PIFE starting. Because readthrough is much more synchronous with decomposing termination than recycling termination, recycling appears to be decided over readthrough much earlier than decomposing.

**Co-present catch-up and stand-by pre-terminational modes.** The holistic assay does not differentiate the catch-up and stand-by pre-terminational pathways. In order to solely monitor the termination that is mediated by ρ in the stand-by mode (called stand-by termination) rather than by ρ in the catch-up mode (called catch-up termination), we pre-incubated ISCs with ρ and washed away the unbound before resuming transcription with NTPs but without adding ρ (Fig. 2a). Although it is not known what fraction of ISCs is bound to ρ under the conditions of this stand-by single-molecule assay, only pre-bound ρ in the stand-by mode can mediate termination because unbound ρ for catch-up mode is washed away. TE increases as more ρ is input in the pre-

incubation (Fig. 2b) demonstrating the stand-by ρ-dependent termination[9].

Interestingly, almost all stand-by termination events are followed by decomposing outcome (Supplementary Fig. 5a) and very few by recycling outcome (Fig. 2b). Because recycling is virtually null throughout a wide range of ρ concentration, the decomposing data overlap with their sum data (not shown) in Fig. 2b. These results indicate that RNAP dissociates off DNA together with RNA release at stand-by termination, and few remain on DNA after stand-by termination.

With stand-by ρ, decomposing termination takes place much later (485 s) than readthrough (207 s) as shown in Fig. 2c. As the time interval between ISC washing and NTP injection prolongs, the stand-by TE gradually decreases with a time constant of 693 s (Supplementary Fig. 6), indicating that the activity lifetime of ρ·RNAP complex is finite but long enough to reveal stand-by termination.

Next, in order to solely measure the catch-up termination, we pre-incubated ISCs with inactive $\rho^{P297S}$ mutant[25], washed away the unbound, and resumed transcription by adding NTPs and wild-type ρ (Fig. 3a). In this catch-up single-molecule assay, the stand-by site on RNAP should be stably preoccupied and masked by a fully inactive mutant so that only catch-up wild-type ρ can mediate termination.

For this purpose, three previously characterized ρ-mutants[25], $\rho^{P279S}$, $\rho^{G51V}$, and $\rho^{Y80C}$, were confirmed to be completely inactive exhibiting only background termination (Fig. 1d) and tested for their masking activity in a modified stand-by assay. We pre-incubated ISCs with an inactive ρ mutant, washed away the unbound, added wild-type ρ, washed away the unbound again, and resumed transcription with NTPs alone (Supplementary Fig. 7). When $\rho^{P297S}$ is pre-bound, it is hardly replaced by subsequently added wild-type ρ, exhibiting full masking, whereas $\rho^{G51V}$ and $\rho^{Y80C}$ mutants achieve only partial masking.

While the catch-up assay's experimental setup with pre-bound $\rho^{P297S}$ is validated, both decomposing and recycling terminations are observed with catch-up ρ in our single-molecule assay (Fig. 3b, Supplementary Fig. 5b) in welcome contrast to a previous report that pre-binding of the mutant ρ completely blocks the termination by subsequent wild-type ρ in a bulk assay[9]. With catch-up ρ, recycling termination (21 s) happens much earlier than decomposing termination (189 s) or readthrough (177 s) as shown in Fig. 3c, which is consistent with the results of holistic assays (Fig. 1f).

**The generality of termination heterogeneity with five different terminators.** In order to test the generality of the molecular heterogeneity of *E. coli* ρ-dependent termination observed with *mgtA*

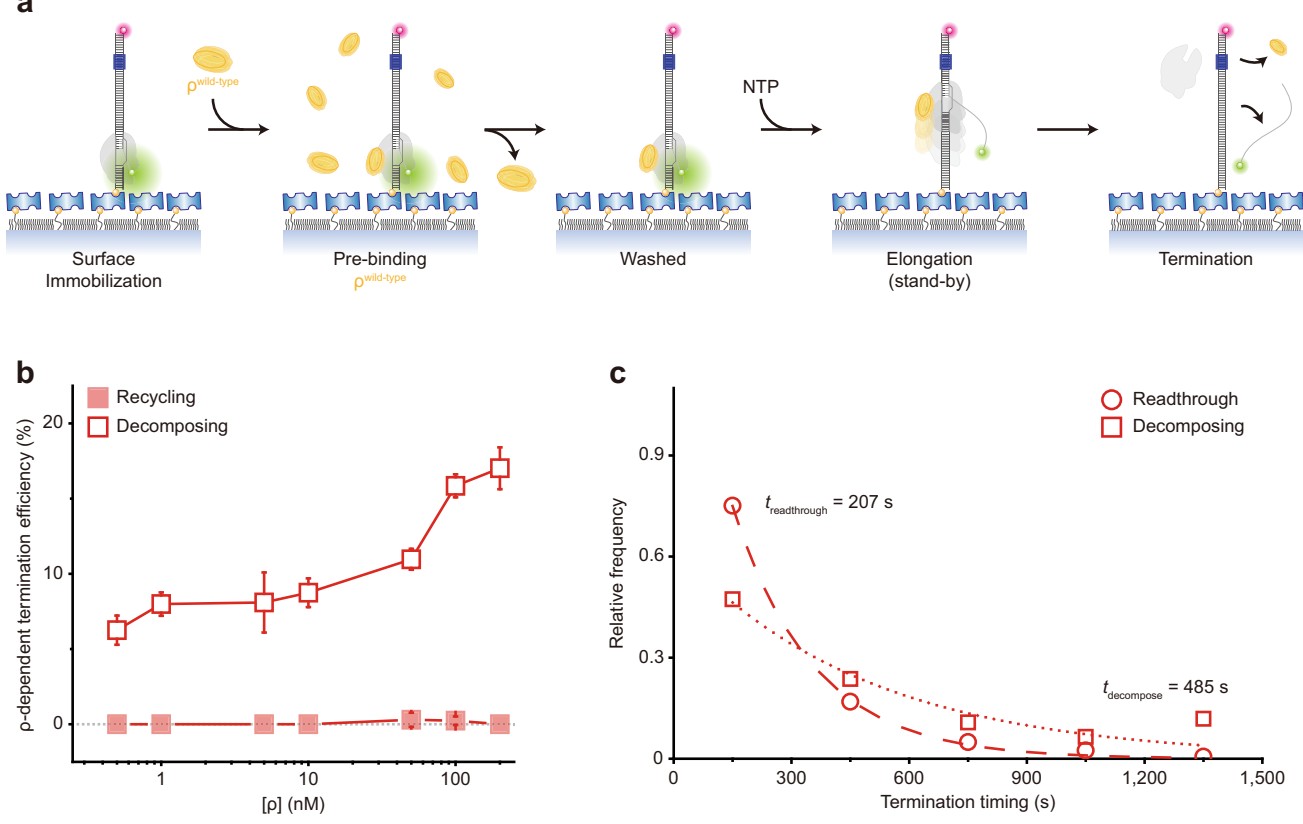

**Fig. 2 Stand-by ρ mediating only decomposing but not recycling termination. a** Scheme for the stand-by single-molecule assay. Termination is mediated only by the stand-by ρ pre-bound on RNAP in the absence of catch-up ρ. **b** ρ-dependency of stand-by decomposing termination. The ρ-dependent TEs of decomposing (open) and recycling (solid) in the y-axis are plotted against ρ concentrations on a base-10 log scale in the x-axis. Recycling is virtually null over the entire range of ρ concentrations used, so the decomposing data overlap with their sum data (not shown). **c** Relative frequencies of readthrough ($n = 326$ complexes) and decomposing ($n = 93$) timings measured in stand-by assays. The data are fitted to single exponential functions. Error bar represents the standard deviation of the mean from $n \geq 3$ independent experiments. The numbers of analyzed molecules for (**b**) are in Supplementary Table 2. Source Data file includes the data for (**b**, **c**).

terminator, we examined four additional ρ-dependent terminators, *rho*, *ribB*, *trp t'*, and *λ tR1* (Supplementary Table 1). Their ρ-dependent termination proficiencies are confirmed in this study by performing in vitro bulk transcription assays (Supplementary Fig. 8), all consistent with the previous reports[7,9,12,18,19,26–30].

The ρ-dependent TEs of the five terminators measured by the single-molecule holistic assays range from 12 to 51% as both catch-up and stand-by TEs vary on terminators (Fig. 4a, Supplementary Fig. 9). These results are consistent with that ρ in either mode interacts specifically with RNA *rut* sequence as well as with RNAP. Furthermore, both catch-up (cyan in Fig. 4a) and stand-by (red) pre-terminational modes are co-present in the operation of every terminator except *λ tR1*, in which stand-by mode is negligible. Accordingly, the pre-terminational mechanism is generally heterogeneous in the operation of a single terminator.

From all the five terminators, the only decomposing outcome is observed in stand-by assays, while both decomposing and recycling outcomes are detected in both catch-up and holistic assays (Fig. 4a). On the other hand, sums of catch-up and stand-by TEs are not always the same as the TE measurements in holistic assays: the sum is smaller than the holistic measurement for *mgtA* and larger for *rho* and *ribB*, and they are similar to each other for *trp t'* and *λ tR1*.

*E. coli* NusA and NusG factors (NusA/G) are known to participate in stable EC along with ρ[7,12]. In order to examine how they influence the termination routes, we added them to ISCs in

all three assays with *mgtA* terminator. Expectedly, the activity lifetime of stand-by ρ is longer with NusA/G than without them (Supplementary Fig. 6), and the background termination little changes. With NusA/G, the catch-up and stand-by modes still coexist and the termination route composition little changes (Fig. 4b).

All the results with and without NusA/G indicate that transcription with ρ proceeds in five different routes; (1) catch-up → readthrough → runoff, (2) catch-up → termination → decomposing, (3) catch-up → termination → recycling, (4) stand-by → readthrough → runoff, and (5) stand-by → termination → decomposing. While the stand-by → termination → recycling path is not operating in any terminator, the operation of a single ρ-dependent terminator runs via the three routes starting with catch-up or stand-by ρ.

When portions of the three termination routes in all termination events from each terminator were separately calculated (Fig. 4c), the catch-up → decomposing route (blank blue box) is the most frequent termination with all terminators, while relative frequencies of the other two terminations depend on terminators. The stand-by → decomposing route (blank red box) is more frequent than the catch-up → recycling route (solid blue box) in three terminators but less in the other two.

Next, the measured timings of recycling termination, decomposing termination, and readthrough events were compared among each other, while the five terminators without NusA/G and *mgtA* terminator with NusA/G were grouped together in a

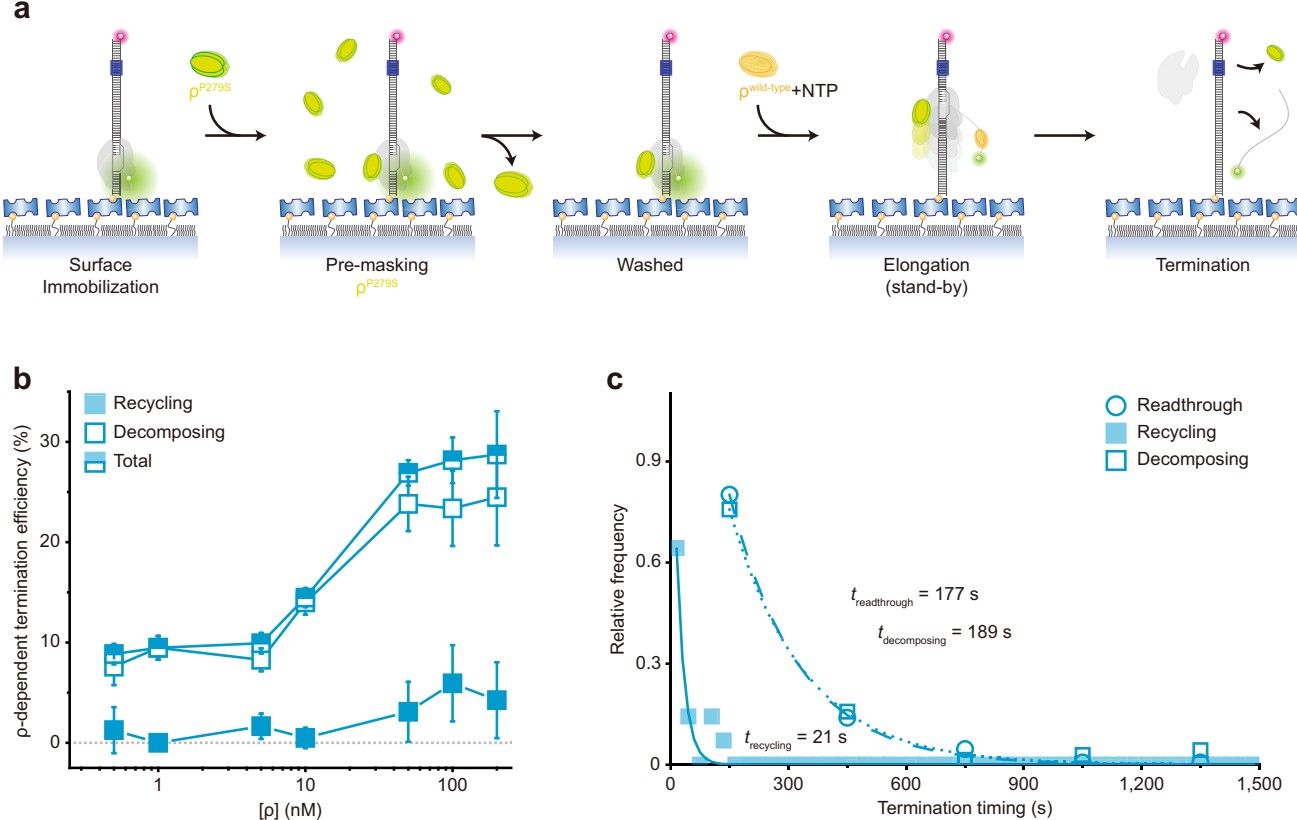

**Fig. 3 Catch-up ρ mediating both decomposing and recycling terminations. a** Scheme for the catch-up single-molecule assay. Termination is mediated only by catch-up ρ, while an inactive ρ pre-occupies the stand-by site on RNAP and is not replaced by subsequently added wild-type ρ. **b** ρ-dependency of catch-up recycling and decomposing terminations. The ρ-dependent TEs of recycling (solid) and decomposing (open) in the y-axis are plotted against ρ concentrations on a base-10 log scale in the x-axis. **c** Relative frequencies of readthrough ($n = 251$ complexes), recycling ($n = 14$), and decomposing ($n = 70$) timings measured in catch-up assays. The data are fitted to single exponential functions. Error bar represents the standard deviation of the mean from $n \geq 3$ independent experiments. The numbers of analyzed molecules for **b** are in Supplementary Table 2. Source Data file includes the data for (**b**, **c**).

box plot with median and quartile values (Fig. 4d). According to student t-tests, the catch-up recycling termination (route 3) is the earliest and especially earlier than the catch-up readthrough (route 1) ($P = 0.001$). Decomposing termination is observed later than readthrough by catch-up ($P = 0.03$) or stand-by ρ ($P = 0.01$).

These chronological orders are in fact more evident when comparisons are made directly between termination and read-through timings for each individual terminator (Fig. 4e). In the graphs of holistic (left), stand-by (center), and catch-up (right) assays, all open squares are located above the diagonal guideline, indicating that decomposing termination is always later than readthrough in all terminators with catch-up or stand-by ρ. In stark contrast, all solid squares are located below the guideline in the catch-up and holistic assay graphs, indicating that recycling termination is always earlier than readthrough with catch-up ρ in every terminator.

Furthermore, decomposing results later with stand-by ρ than catch-up ρ or almost at the same time in all terminators, when the decomposing termination timings by the two ρ modes are directly compared from each other in individual terminators as shown in Fig. 4f, while it is not revealed in box-plot comparisons of Fig. 4d. These results suggest that stand-by ρ tends to take more time or energy to decompose EC than catch-up ρ. It is interesting that stand-by ρ would encounter RNAP earlier than catch-up ρ but tend to achieve termination not so much earlier than catch-up ρ.

**RNA shearing and RNAP displacing for terminational release of RNA.** Finally, we examined whether RNA shearing or RNAP hyper-translocation is utilized in ρ-dependent termination. In the RNA shearing model[14], ρ pulls RNA to resolve RNA·DNA hybrid for terminational release of RNA and immediate or delayed dissociation of RNAP from DNA. In the RNAP hyper-translocation model[14], by contrast, ρ pushes RNAP forward on DNA for the collapse of transcription bubble leading to RNA release with simultaneous or subsequent RNAP dissociation.

For testing the RNA shearing model, we designed a DNA scaffold with the *mgtA* terminator mutant AU100 where AU content of the RNA·DNA hybrid formed at the termination site is increased to 100% from the wild-type 44% (Fig. 5a, Supplementary Table 1). Thus, AU100 would form a weaker hybrid than the wild-type while pausing at the termination site. The weaker hybrid in AU100 would facilitate the termination by RNA shearing, while it would not bring much energetic advantage to RNAP hyper-translocation. In this test used was *mgtA* terminator exhibiting a single major termination site[18,19] (Supplementary Fig. 8).

Recycling TE is roughly doubled with AU100 in Fig. 5b, as ρ-dependent TEs (= raw TE—background TE) are normalized to the maximum of 100% by dividing them with (100%—background TE), when black solid parts are compared between the wild-type and AU100 in holistic assays or cyan solid parts in catch-up assays. Including four more terminator mutants (Supplementary Table 1), a total of six templates with varying AU contents together exhibit that recycling TE increases with

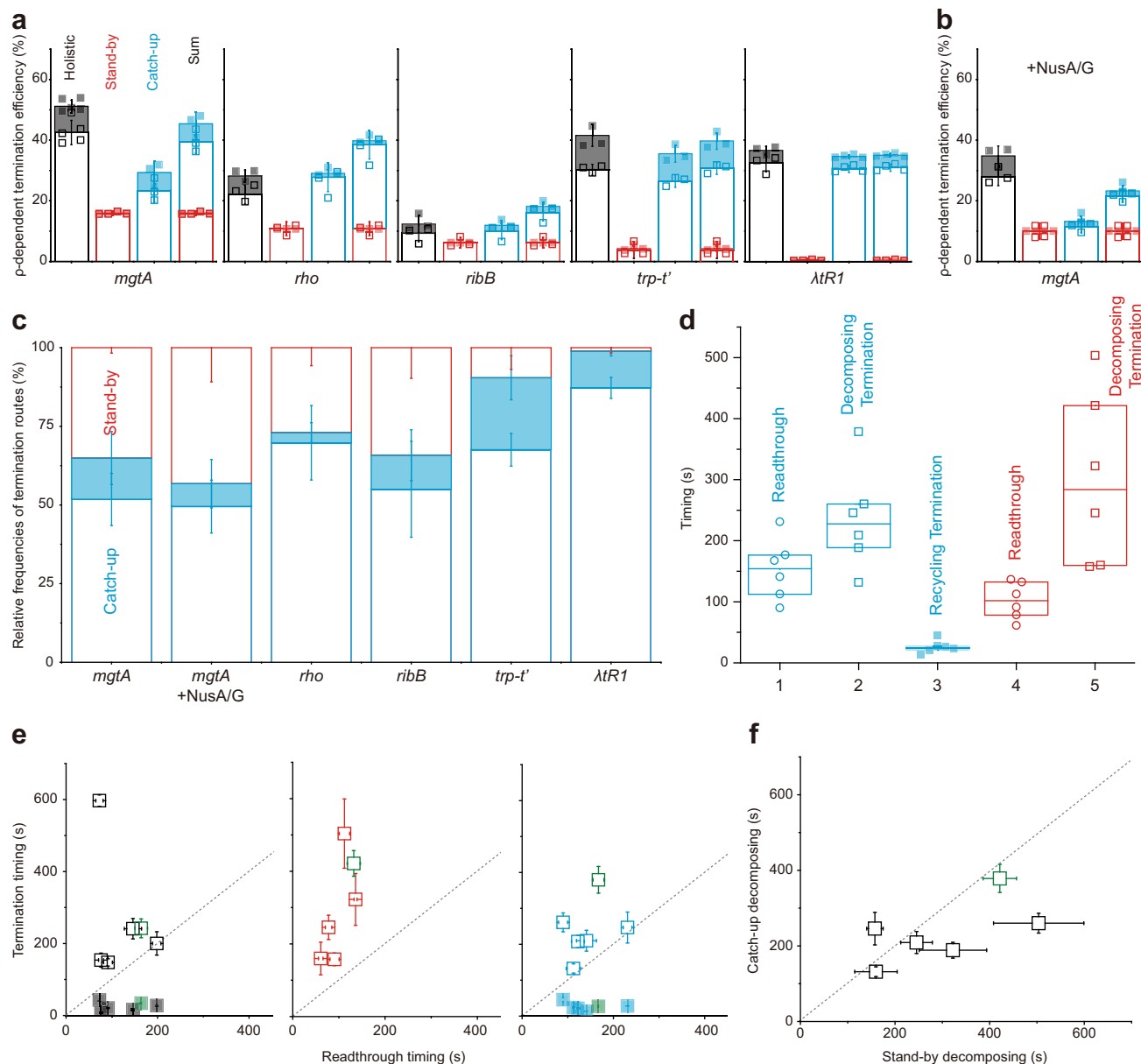

**Fig. 4 Generality of ρ-dependent termination route heterogeneity.** Recycling (solid) and decomposing (open) terminations were counted and timed with each of the five terminators in holistic (black), stand-by (red), and catch-up (cyan) assays. **a** ρ-dependent TEs measured in the three different assays. The ρ-dependent TEs in catch-up and stand-by assays are summed in the rightmost bars for comparison with the holistic measurements in the leftmost bars. **b** The same as in **a** but for *mgtA* terminator with NusA/G. **c** Relative frequencies of the three termination routes. With each terminator, the frequency of each termination route in total termination events was estimated using the data in (**a**) and (**b**). The catch-up → decomposing route is the most frequent with every terminator. **d** Relative timings of readthrough and termination. For each transcription route with ρ, a box plot is drawn with readthrough or termination timings observed in all terminators in (**a**) and (**b**) ($n = 6$), their median (centerline), and the first and third quartiles (boundaries). When the timing distributions are compared among the five transcription routes, the catch-up ρ's recycling termination is the earliest. **e** Comparison between termination and readthrough timings. The average termination timings of all terminators in the *y*-axis are plotted against the average readthrough timings in the *x*-axis. The data of *mgtA* with NusA/G are colored green. All open squares (decomposing termination) are located above the diagonal eye-guide line with a unit slope (dotted), and all solid squares (recycling termination) below the guideline. **f** Comparison of decomposing termination timings between catch-up and stand-by modes. The average timings of decomposing termination by catch-up ρ in the *y*-axis are plotted against those by stand-by ρ in the *x*-axis. All open squares are located below the guideline, except for *rho* terminator slightly above the line. Error bar represents the standard deviation of the mean from $n \geq 3$ independent experiments. The numbers of analyzed molecules for (**a**) and (**b**) are in Supplementary Table 2. Values and statistics of timings for **d**–**f** are in Supplementary Tables 3 and 4. Source Data file includes the data for (**a**–**f**).

rising AU content (Fig. 5c), while decomposing TE is little correlated with it (Fig. 5d). These results support that RNA shearing results in recycling on DNA much more likely than decomposing outcome, i.e., RNA release is followed by much-delayed dissociation of RNAP from DNA.

The uncorrelated variations in Fig. 5d are so large that some mechanism(s) different from RNA shearing can be presumed for decomposing termination, so the hyper-translocation model was tested (Fig. 6a) as previously demonstrated using modified DNA scaffolds where 3-bp mismatch-causing mutations are introduced

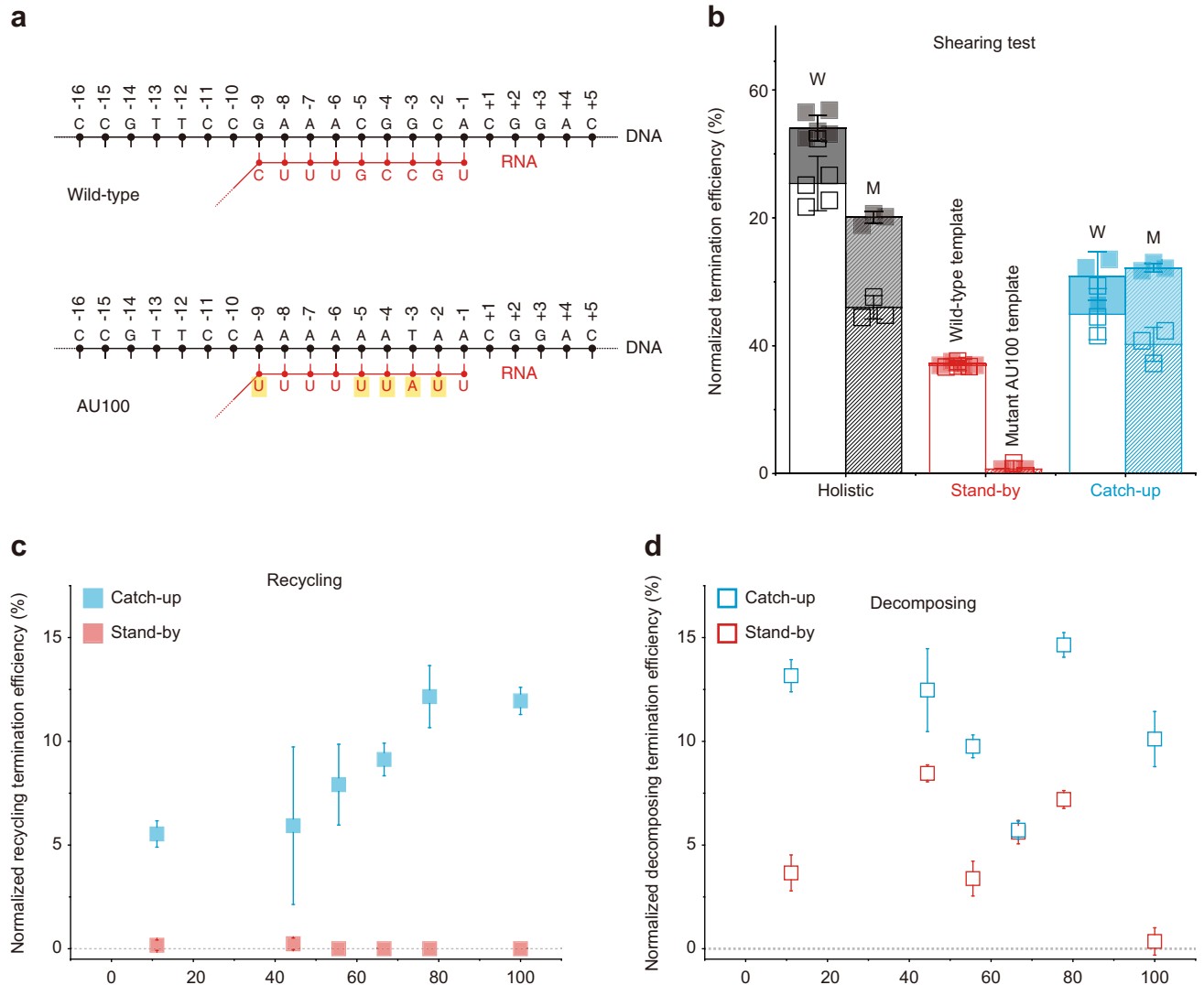

**Fig. 5 RNA shearing resulting only in RNAP recycling on DNA.** For RNA shearing tests, comparisons were made among six templates producing RNA·DNA hybrids with varying AU contents. ρ-dependent TEs (= raw TE—background TE) of recycling (solid) and decomposing (open) were measured in holistic (black), stand-by (red), and catch-up (cyan) assays and normalized to the maximum of 100% by dividing with (1—background TE). **a** Putative RNA·DNA hybrids formed at the termination site. The AU content is 44% in the wild-type *mgtA* and 100% in AU100 mutant. **b** Augment of recycling termination by hybrid weakening in AU100. Normalized ρ-dependent TEs are compared between the wild-type (W) and AU100 (M). **c** Positive correlation of recycling termination with hybrid instability. Normalized recycling TEs in the *y*-axis are plotted against the AU content in the *x*-axis. **d** No correlation of decomposing TE with hybrid instability. Normalized decomposing TEs in the *y*-axis are plotted against the AU content in the *x*-axis. Error bar represents the standard deviation of the mean from $n \geq 3$ independent experiments. The numbers of analyzed molecules for (**c**) and (**d**) are in Supplementary Table 2. Source Data file includes the data for (**c**) and (**d**).

in the nontemplate strand of template DNA at varying positions[14]. If a mismatch overlaps with a transcription bubble region that is not base-paired before termination but becomes base-paired after hyper-translocation for termination, the mismatch would inhibit rewinding to diminish hyper-translocation proficiency but little affect RNA shearing, which does not need rewinding of the region[14,31].

Using a series of 15 template mutants with a 3-bp mismatch at varying positions (Supplementary Table 1), we observed that the mismatches at certain positions upstream of the termination site (denoted as the −1 position) substantially decrease decomposing termination (Fig. 6b) but not recycling termination (Fig. 6c). Unexpectedly, however, the mismatch effect peaks at the −12 position, which is just upstream of the transcription bubble rather than within it according to the EC structures that were reported

for ρ-dependent terminators[7,12,32–37] although not yet for *mgtA* terminator.

Speculatively, the transcription bubble position could be different in *mgtA* terminator, or RNAP backtracking could be involved in ρ-dependent termination, such that the −13 to −11 positions are wholly or partly located within the bubble just before termination. Alternatively, the mismatch effects may not be related to hyper-translocation but with something else. For example, a DNA region around the −12 position interacts with RNAP in the recent X-ray crystal structures[7,16], and such interaction may be hypothetically important for termination.

Our results, however, at least suggest that duplex formation by template rewinding around the −12 position rather than others is critical for decomposing termination regardless of whether the hyper-translocation is involved or not in ρ-dependent

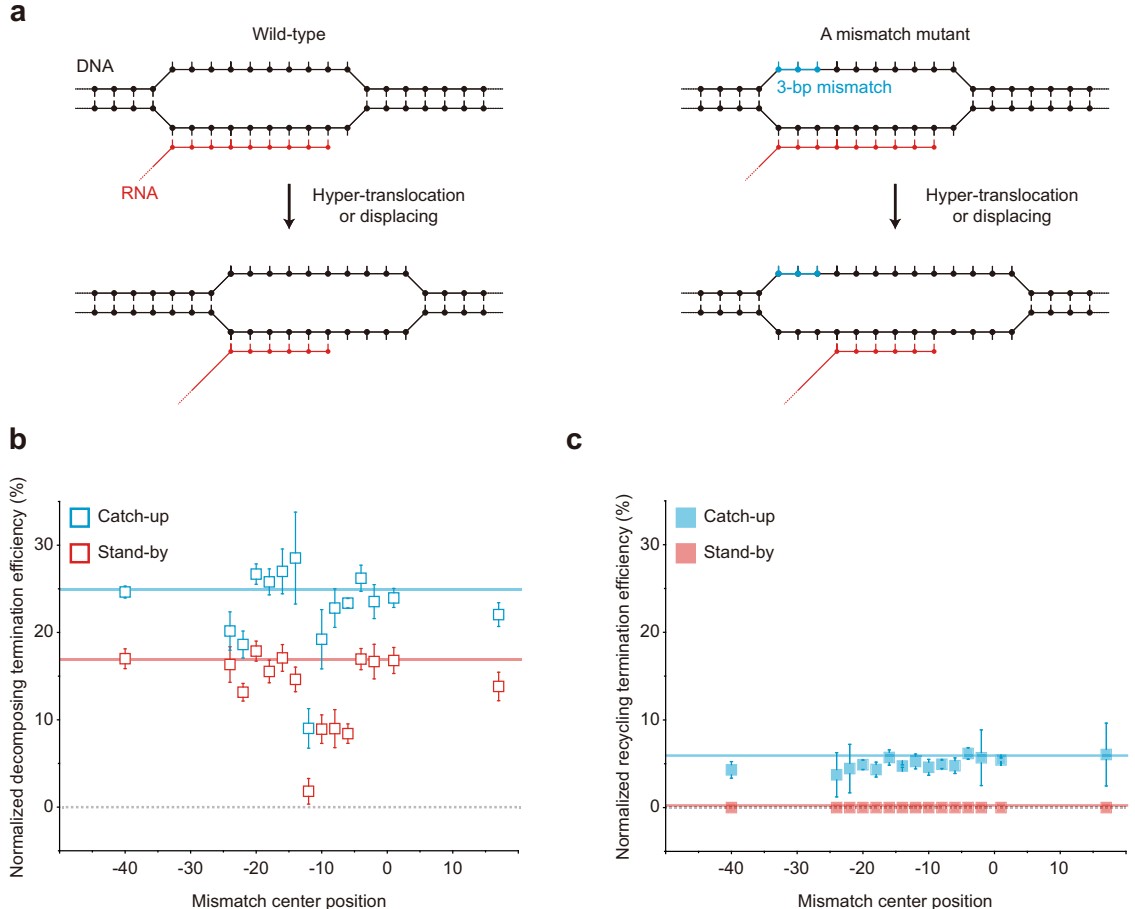

**Fig. 6 RNAP displacing resulting only in EC decomposing.** For RNAP displacing tests, comparisons were made among 15 DNA templates with a 3-bp mismatch. ρ-dependent TEs (= raw TE—background TE) of recycling (solid) and decomposing (open) were measured in stand-by (red) and catch-up (cyan) assays and normalized to the maximum of 100% by dividing with (1—background TE). **a** Putative elongation-coupled progression of transcription bubble in the *mgtA* wild-type template and a mutant with a 3-bp mismatch (blue). **b** A peaked effect of the 3-bp mismatch on decomposing termination. Normalized decomposing TEs of the 15 mutants in the y-axis are plotted against the center position of the 3-bp mismatch in the x-axis. Horizontal lines indicate the values of the wild-type template (16.7% for stand-by assay and 24.9% for catch-up assay). **c** No effect of the mismatch on recycling termination. Normalized recycling TEs in the y-axis are plotted against the mismatch position in the x-axis. Error bar represents the standard deviation of the mean from n ≥ 3 independent experiments. The numbers of analyzed molecules for (**b**) and (**c**) are in Supplementary Table 2. Source Data file includes the data for (**b**) and (**c**).

termination, so it is described here loosely as RNAP displacing instead of hyper-translocation. These results together support that RNAP displacing results in decomposing outcomes much more readily than recycling on DNA, i.e., RNA release is accompanied by fairly simultaneous dissociation of RNAP from DNA.

## Discussion

Mechanisms of ρ-dependent termination have long been studied but still remain elusive. A prevailing debate has been concerned with whether ρ first binds RNA and catches up with RNAP later[13,38] (catch-up mode) or ρ first binds RNAP and stands by for RNA emergence[9] (stand-by mode). Contrary to the previous reports that only one pre-terminational mode is present, we discover that both catch-up and stand-by modes are used by a single terminator. Their proportions, however, vary in different terminators, raising interesting questions of how their proportions are determined and influence gene regulation.

We additionally discover that decomposing outcome is much more frequent than recycling from each of the five tested ρ-dependent terminators, as opposed to that recycling outcome is much more frequent than decomposing from intrinsic

terminators[15–17]. Frequently with catch-up or stand-by ρ, RNA release and RNAP dissociation from DNA occur together, as RNAP displacing results only in EC decomposing. By contrast, less frequently with catch-up ρ, only RNA is released but RNAP is not concomitantly dissociated from DNA when RNA shearing results only in RNAP recycling on DNA, while this is much more frequent in intrinsic termination without ρ.

Our kinetic analyses additionally reveal how these co-occurring mechanisms are chronologically arranged in the course of ρ-dependent termination (Fig. 7). The choice for recycling termination is always made earlier than decomposing termination in all tested terminators. Recycling termination results from RNA shearing and observed only in our catch-up single-molecule assays but not in stand-by assays. Therefore, we suspect that catch-up ρ moving on RNA can shear off RNA much more handily than stand-by ρ residing on RNAP, and pulling RNA alone from EC may need less force than pushing RNAP or EC to be displaced.

Transcriptional readthrough naturally synchronizes with the later termination, which is decomposing termination much more likely than recycling termination. However, decomposing termination completes even later than readthrough, regardless of

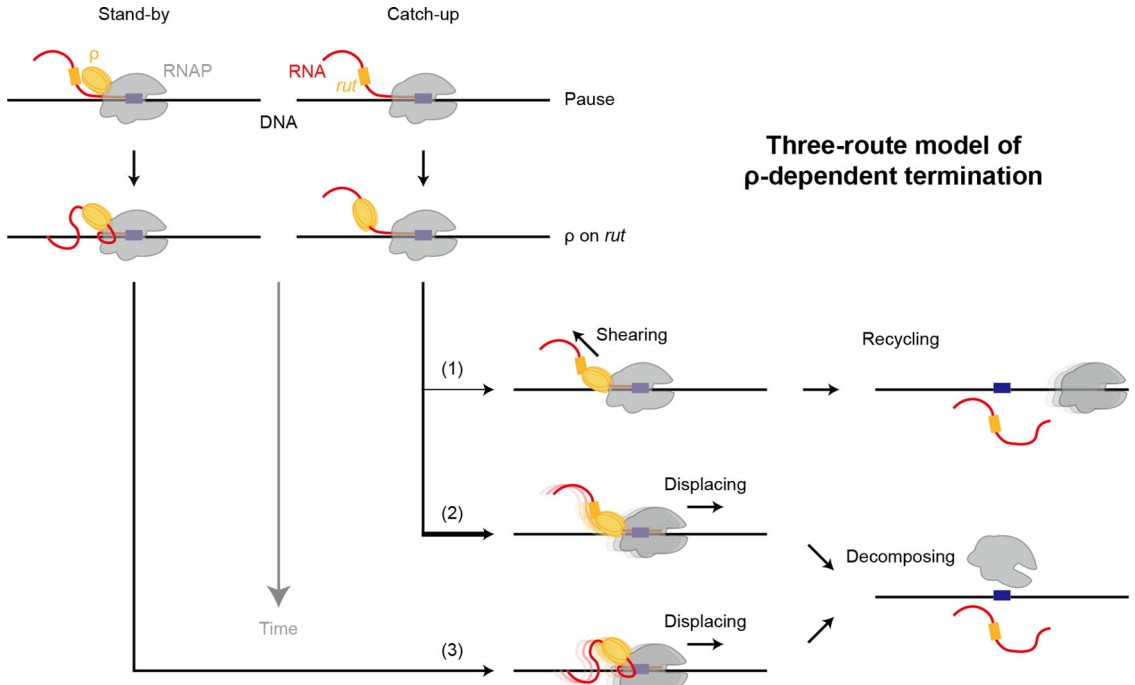

**Fig. 7 The three-route model of ρ-dependent termination.** Three kinetically different routes of transcription termination are operating with a single terminator in our model for ρ-dependent termination, whereas all previous models assume a single path. (1) Catch-up recycling termination. The fastest termination route is taken by catch-up ρ, which binds nascent RNA at a *rut* site, translocates down on RNA, and catches up with RNAP that is pausing at a termination site. Only RNA, not RNAP, is released, and post-terminational RNAP recycles on DNA for 1D reinitiation (like in most intrinsic termination[15–17], which is ρ-independent but RNA hairpin-dependent). (2) Catch-up decomposing termination. If the first, fastest termination fails, the second termination route can be followed by catch-up ρ. Transcription complex decomposes in a step as both RNA and RNAP depart DNA. This second-route termination by catch-up ρ for decomposing always occurs much later than the first-route termination for recycling. (3) Stand-by decomposing termination. The slowest termination route is usually pursued by stand-by ρ, which pre-binds RNAP, moves along DNA with RNAP, and stands by for binding a *rut* site emerging out of RNAP. This termination results in one-step disassembly of the transcription complex. Interestingly, stand-by ρ presumably binds RNAP earlier than catch-up ρ but tends to achieve termination not so much earlier. While the catch-up decomposing termination route (2) is the most frequent in all terminators, relative frequencies of the other routes (1) and (3) depend on terminators.

whether it is mediated by catch-up or stand-by ρ (Fig. 4e). It is possibly because EC decomposing requires more massive or slower conformational changes than readthrough, or less likely RNA temporarily remains on DNA after RNAP displacing.

A simple speculative course of ρ-dependent termination (Fig. 7) could start with the catch-up ρ shears off RNA for recycling termination using less energy. If it fails, the catch-up ρ·EC complex undergoes conformational changes using more accumulated energy for decomposing termination. Lastly, stand-by ρ can additionally mediate the same decomposing termination often more slowly than catch-up ρ. Interestingly, stand-by ρ could bind RNAP earlier than catch-up ρ but does not achieve the same termination more readily.

The catch-up and stand-by pre-terminational modes would follow separate paths on different timescales and could build disparate structures of termination-proficient complex favoring specific outcomes such as RNAP recycling on DNA and EC decomposing at once. We speculate that even for the same decomposing termination, the catch-up and stand-by ρ's could induce dissimilar changes in a termination complex structure because their termination timings often differ much from each other (Fig. 4f).

Most, if not all, previous models of ρ-dependent termination simulate with a single ρ molecule. However, it should not be ruled out that two ρ molecules participate side by side in termination and compete or collaborate with each other. Because the sums of stand-by and catch-up TEs are larger or smaller than the holistic assay measurements (Fig. 4a, b), one can speculate a possibility that the interaction between the two ρ molecules in different

modes is synergistic, antagonistic, or null in a terminator-dependent fashion. Furthermore, the stand-by ρ acting later could offer a standby backup in case the earlier-acting catch-up ρ fails to mediate termination in the same complex.

Incidentally, one can consider it odd that the background termination occurring without ρ results solely in the decomposing outcome, while ρ-independent, intrinsic termination results mostly in recycling[15–17]. However, the decomposing outcome is not absent in intrinsic termination requiring RNA hairpin formation, so one can hypothesize that the background termination results from infrequent decomposing of EC unstabilized without ρ or RNA hairpin, and that shearing RNA alone for recycling occurs actively rather than passively in both intrinsic and ρ-dependent terminations. Apparently, ρ plays the same role as the terminator RNA hairpin to mediate recycling termination and a different role to mediate decomposing termination.

In this study, we show that ρ follows catch-up or stand-by pre-terminational setting, executes RNA-shearing or RNAP-displacing terminational release, and then results in RNAP-recycling or EC-decomposing post-terminational outcome. These mechanisms in the three sequential steps are rather tightly coupled so that only three routes are co-present out of eight ($2^3$) possible stepwise combinations in the performance of a single terminator. The three routes apparently operate on individual timescales and lead to their specific outcomes. This three-route model (Fig. 7) could provide a platform to review the prior studies on ρ-dependent termination and to investigate further compelling aspects of transcription regulation.

## Methods

**Single-molecule experiments of transcription termination.** Individual transcription complexes were immobilized to quartz microscope slides via biotin-streptavidin conjugation on the slide surfaces that were coated with polyethylene glycol (PEG) and biotin-PEG5000 (Laysan Bio) in 1:40 ratio[39] and then with streptavidin (0.2 g/l, Invitrogen) for 5 min. The complexes were then imaged under a homemade wide-field total-internal-reflection fluorescence microscope equipped with 532-nm green (Excelsior-532-50-CDRH) and 640-nm red (Excelsior-640c-35) lasers (Spectra-Physics) using an electron-multiplying charge-coupled camera (iXon DU-897, Andor Technology). All experiments were performed at 37 °C with 1-s exposure time in alternating laser excitation mode[40] and time resolution was 2 s. Softwares IDL 7.0 (ITT), MATLAB R2018a (MathWorks), and Origin 8.5 (OriginLab) were used for data analysis.

ISCs were prepared by incubation of DNA, Cy3-labeled ApU (250 µM, TriLink BioTechnologies), ATP, CTP (10 µM each, GE Healthcare), and RNAP holoenzyme (340 nM) in a transcription stalling buffer (20 mM Tris-HCl, pH 8.0, 20 mM MgCl$_2$, 20 mM NaCl, and 1 mM dithiothreitol) for 20 min. Transcription was resumed with all four NTPs (200 µM each) and ρ (100 nM or varying) in a transcription resuming buffer (40 mM Tris-HCl, pH 8.0, 10 mM MgCl$_2$, 150 mM KCl, 1 mM dithiothreitol, 5 mM protocatechuate acid, 100 mM protocatechuate-3,4-dioxygenase, and saturated Trolox). When needed, NusA (500 nM) plus NusG (500 nM) or a ρ mutant (100 nM) was added to the stalling buffer.

**Preparation of *E. coli* proteins.** RNAP holoenzyme was either purchased from New England Biolab (M0551S) or custom-purified as previously described[41,42]. ρ factor was purchased from Bioprogen. Three ρ mutants were expressed and purified as previously described[25] (Supplementary Fig. 10) using their expression plasmids that were provided as a gift by Ranjan Sen in Hyderabad, India. NusA was expressed with an N-terminal His$_6$ tag from a cloned plasmid pNG5 in *E. coli* BL21(DE3) and purified by using HiTrap IMAC HP affinity and Superdex 200 gel filtration[35]. NusG was expressed with a C-terminal His$_6$ tag from a cloned plasmid pRM1160 in *E. coli* BL21(DE3) and purified by using HiTrap IMAC, HiTrap Q, and HiLoad Superdex 75 columns[32].

**In vitro bulk transcription assay.** *E. coli* RNAP holoenzyme (30 nM) and a template DNA (15 nM) with one of the five terminators were incubated in a transcription buffer (20 mM Tris-HCl, pH 8.0, 10 mM MgCl$_2$, 100 mM KCl, and 1 mM dithiothreitol) at 37 °C for 10 min. The incubation was continued with ρ at 0, 15, 30, or 60 nM and NusA/G each at 0 or 50 nM for 10 min, and then with 200 µM ATP, 200 µM GTP, 200 µM UTP, 0.025 µM CTP, and 0.05 µM [α-$^{32}$P]CTP for another 10 min. The transcription reaction was quenched by the addition of a 2× urea-denaturing gel loading buffer and analyzed on a 6%-polyacrylamide urea gel. While raw TE is the ratio of termination band intensity to the sum of termination and readthrough band intensities, ρ-dependent TEs are estimated by subtraction of the background TE measured without ρ from the raw TEs measured with ρ.

**Preparation of transcription templates.** Using DNA oligomers (Supplementary Table 1) purchased from Integrated DNA Technologies, nontemplate DNA strands were prepared by annealing their two parts with a DNA splint by slowly cooling from 90 to 16 °C in 50 mM Tris-HCl, pH 8.0, 10 mM MgCl$_2$, 10 mM dithiothreitol, and 1 mM ATP, and by ligating them using T4 DNA ligase 2 (New England Biolabs). Homo-duplex DNAs were made by polymerase chain reactions (PCRs) using biotin-labeled forward and Cy5-labeled backward primers. Hetero-duplex DNAs were made by PCRs using biotin-labeled forward and unlabeled backward primers for wild-type DNAs, and using unlabeled forward and Cy5-labeled backward primers for mutant DNAs. Then, PCR products were annealed, and products with both biotin and Cy5 were used in experiments.

**Reporting summary.** Further information on research design is available in the Nature Research Reporting Summary linked to this article.

## Data availability

Source data are provided in a Source Data file. All other relevant data are available from the corresponding authors upon request. A reporting summary for this Article is available as a Supplementary Information file. Source data are provided with this paper.

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

## Acknowledgements

We thank Prof. Kook Sun Ha for the initial design and construction of fluorescent transcription complexes, and Dr. Ranjan Sen for providing expression plasmids of the three ρ mutants. This work was supported by grants from the National Research Foundation of Korea (2019R1A2C2005209 to S. Hohng, 2019M3E5D6066058 to J.Y.K., and 2018R1A2B2004602 to Y.S.) and from the High-Risk High-Return Project of KAIST (N10110078 to C.K.).

## Author contributions

S. Hohng, E.S., H.U., C.K., and J.Y.K. conceived the study. E.S and H.U. designed and performed all biophysical experiments by measuring single-molecule fluorescence images. P.R.M. and Y.S. prepared ρ mutants and performed all biochemical experiments. J.Y.K. and S. Hwang prepared *E. coli* RNAP, NusA, and NusG and performed bulk transcription assays. C.K., J.Y.K., and Y.S. supervised the molecular biology and biochemistry parts. S. Hohng supervised the biophysics part. E.S, S. Hohng, C.K., and J.Y.K. collectively analyzed all the data. All authors contributed to writing and revising the paper.

## Competing interests

The authors declare no competing interests.
