## [Peer Review File · Nature Communications]

Reviewers' Comments:

Reviewer #1:

Remarks to the Author:

In this manuscript by Song et al., the authors use single-molecule fluorescence to observe extrinsic transcription termination by rho helicase. Their aim is to determine whether rho accompanies RNAP (stand-by termination) or loads onto RNA at a rut site (catch-up termination), and furthermore to determine whether these modes of termination occur via hypertranslocation of RNAP relative to DNA, or via shearing of RNA out of the RNAP. By attaching a Cy3 dye to the 5' end of the nascent RNA, and a Cy5 dye at the free DNA end, the authors can distinguish several events. They can observe RNAP begin to transcribe (by a decrease in Cy3-PIFE), and conduct either « recycling » termination (by a loss of Cy3 signal representing loss of RNA and a gain of Cy5-PIFE representing RNAP at the DNA end), « readthrough » (when Cy3 signal does not disappear and a Cy5-PIFE signal appears), and « decomposing » termination (by loss of Cy3 signal, which represents loss of RNA, with no simultaneous appearance of Cy5-PIFE). The relative fractions of these processes are then quantified and compared in situations where rho is pre-bound to RNAP (stand-by termination) or where rho has to move from the rut site to RNAP to remove it (catch-up termination). The main conclusion here is that both stand-by and catch-up termination coexist, and that catch-up termination leads to either decomposing or recycling termination (in a 5:1 ratio) whereas standby termination leads to decomposing but not recycling termination. Because recycling termination is enhanced when the AU content of the extrinsic terminator is increased, and because this change is expected to affect shearing but not hypertranslocation termination, the authors conclude that recycling termination occurs via shearing. Similarly the authors try to determine if decomposition occurs via hypertranslocation by measuring the effect of a 3-bp mismatch in the DNA:DNA hybrid, on the idea that a mismatch will inhibit duplex reformation and thus hypertranslocation but not shearing, however here the results do not conform to expectations and lead the authors to speculate on novel features of termination processes and RNAP structures.

Overall the authors convince this reviewer that both stand-by and catch-up activities of rho take place, and that when stand-by termination is blocked by prebinding a catalytically dead rho to RNAP, catch-up termination can still take place. Finally, RNAP recycling via rho-dependent catch-up termination appears to take place via shearing and catch-up termination can lead to decomposition or recycling but standby termination can lead only to decomposition. At the same time, the authors do not resolve the question of whether classical « decomposing » rho-dependent termination occurs via shearing or hypertranslocation – the looming question in the field as indicated at the start of the manuscript – and because the vast majority of rho-dependent termination events occur via decomposing and not recycling (itself a phenomenon which has only been described recently) this is somewhat unfortunate and reduces the impact of the work. From a technical standpoint, this reviewer is also concerned about the reaction conditions used in this manuscript. In particular, the authors use an anti-bleaching solution based on the PCA/PCD system, which is known to rapidly acidify reaction conditions and lead to non-specific interactions (see Swoboda et al., ACSNano 6 : 6364–6369 (2012). Combined with only 10 mM of Tris buffering agent this seems potentially risky, and a pyranose oxidase antibleach seems like a better choice. How are the authors certain that this is not causing problems in their assay ?

Minor points :

1. Related to the reaction conditions discussed above, it is surprising that the runoff times are so long (on the order of 200 seconds). Given the transcript is only ~200 base-pairs, this would suggest the polymerase is only moving at approximately one base per second. This is at least a factor of ten slower than what is expected. Perhaps this is related to the pause at the terminator, but perhaps the reaction conditions are far from optimal ? Have the authors carried out measurements for the runoff time in the absence of the terminator to show that their RNAP is working normally ?
2. Is it possible that termination/recycling is undersampled, for instance if the polymerase diffuses upstream (where there is no Cy5) rather than downstream of the terminator ? How can the authors rule this out ?

3. How long does Cy5 PIFE last for in runoff – does the polymerase stay stuck indefinitely to the DNA end once it reaches it ?

4. It is a bit difficult to see the low concentration rho points in the termination efficiency graphs, and it is therefore a bit hard to see how standby and catch-up sum to the holistic assay.

5. The authors should better specify the number of events they are tallying and analyzing. They state that most data are from three experiments on over fifty molecules, but are multiple events measured on the same molecules or is there only one event from each molecule for instance ?

Reviewer #2:

Remarks to the Author:

The authors present the results of their single molecule fluorescence-based assay looking at the pathway (or pathways as it turns out) of rho-dependent termination. Their data provide evidence that multiple distinct outcomes can be observed and may help unify previous reports supporting one of various models. These models principally RNA-centric or RNAP-centric recruitment of Rho and whether the polymerase dissociates or remains bound to the DNA template. The work nicely points out that molecular mechanisms don't necessarily only proceed on a linear pathway and also goes further to show that the fraction of individual outcomes can change depending on template sequence. The work is of interest to the field and provides a unique view into the molecular processes underlying termination in bacteria. I would recommend publication subject to my major points below:

Major points

(1) The aspect of pausing during termination is not adequately discussed or analyzed. However, the pause time is a crucial step which gives the time needed for termination to occur. The first measurement that should be made is a measurement of the pause time in the presence and absence of rho in both the "catch-up" and "stand-by" assays. This can be done by comparing a DNA template lacking the termination sequence to the templates already described. Does the presence of Rho in stand-by mode change the pause kinetics? If not, then the pause part of the mechanism can be placed upstream of any Rho-dependent bifurcation of mechanism. This will allow the authors to dissect the times they measure for termination in the different situations on top of the sequence dependent termination pause.

(2) Controls in the absence of a promoter sequence and/or in the absence of sigma factor need to be performed to show that the authors are observing truly promoter-dependent transcription complexes.

(3) In Fig. 2B, a plot of the total termination efficiency along with the fractional plots would provide a clearer picture of what is going on.

(4) Are the times reported as to when termination occurs from when NTPs are flowed? Should these times be thought of as the sum of the termination pause + whatever other molecular transitions stimulated by Rho? Estimates or, even better, measurements of initiation time, elongation time, and pause times will provide context for these measurements. These times also seem quite long. Are they commensurate with other measurements of the kinetics of termination?

(5) Extended data Fig. 5 shows that masking rho mutations suppress the full stand-by signal, but it is a small effect. How are the error bars calculated in this figure? They seem quite small! How many observations have been made under each condition? This comment applies to other figures as well. The number of observations need to be indicated.

(6) I want to commend the authors on performing their assay on multiple terminators and pointing out that outcomes change!!! This is great!

(7) I don't think one should use termination efficiency (normalized TE is probably even worse) as a measure of binding as the efficiency could change within the context of a bound complex.

(8) What are the implications of the distinct timings reported? Presumably, these mechanisms will be engaged in a sort of kinetic competition with each other. In the cellular environment, does one expect there to always be a rho in stand-by mode based on concentrations?

With the stand-by rho positioned and extra added at NTP chase, shouldn't we expect stand-by to dominate? Why is it slower than "catch up"?

Minor points

(1) To what degree are Cy3/Cy5 signals co-localized to assure that the Cy3 signal is DNA-dependent?

(2) Caption in ED Fig. 6 says, "background efficiencies were measured with rho", but shouldn't this read, "without rho"?

Reviewer #3:

Remarks to the Author:

The manuscript entitled, "Molecular.....termination", by Song et al., describes in vitro single-molecule measurements of Rho-dependent transcription termination process using an immobilized DNA template on a glass slide following the fluorescence signals emanating from the probes tagged at the DNA and the nascent RNA ends. The authors attempted to incorporate different heterogeneous behavior of the transcription machinery on the slide surface into different transcription termination models of Rho-dependent termination. I have many serious concerns about the experiments and their interpretations.

General comments:

1. Many experiments have serious flaws in design and lack proper controls (see below under specific comments). To me, data were over-interpreted to fit into some mechanistic models.
2. Except for λ tR1 and trpt', all the other terminators used are not properly characterized in vitro. Hence, regular in vitro transcription termination assays in bulk using radio-labeled NTPs is necessary to identify the termination sites, the efficiency of termination, termination kinetics, etc., to validate the observations and interpretations of the single-molecule experiments.
3. The authors unnecessarily coined new terminologies like, catch-up, stand-by, holistic assays, etc that complicated the descriptions. Terminologies like "RNA-dependent", "RNAP-dependent pathways", "kinetic coupling" are well established in the field and the literature. There is no need for new jargon.
4. The overall descriptions of different experiments are too sketchy and were very difficult to comprehend in many places. A thorough re-writing is required.

Specific comments:

Figure 1:

- i) Show in vitro transcription data in solution with all the templates used in the study to identify the termination sites and to check whether they are matching with the fluorescence signals obtained in the single-molecule studies. Rho terminates over a wide range at multiple sites, which are needed to define for the uncharacterized terminators like, mgtA, rho, and rib. Heterogeneity observed in the single-molecule assays could be because of that. Speed of RNA or RNAP release could depend on the sequence and pausing states at each of these sites. Multiple termination events do not mean the existence of different mechanisms.
- ii) A control transcription assay without Rho is needed.
- iii) No statistics have been provided describing the number of molecules that exist in each type of event.
- iv) Two events occur during the termination process, RNA release and RNAP release. There is no need to use terminology like "decomposition". Slow-release of RNA coincides with the slow release of RNAP in 3rd type (1C) of event, which could also mean two events occur simultaneously. As

there are multiple termination sites, why step-wise disappearance of Cy3-signal was not observed? The transcription elongation process itself is heterogeneous, which is reflected in these assays.

v) The PIFE signal went down after some time, which means RNAP may not be recycled immediately. To prove the recycling of RNAP, the second round of transcription events with the same RNAP has to be shown experimentally.

vi) From how far the PIFE could be observed? Can RNAP stalled/paused at the termination sites induce PIFE? These needed to be clarified before interpreting the data.

vii) As there are multiple termination sites, depending on the release kinetics of RNA and RNAP at each of these sites, don't we expect to see these different types of events at different sites?

In general, ECs are highly heterogeneous under in vitro conditions, so heterogeneity at termination sites is expected. Some ECs are refractory to termination, some are slow releasing and some are fast releasing type. Hence, there is no need to introduce new terminologies like recycling, decomposition, holistic assays, etc, which confuses the reader.

I am not at all convinced that these data support the existence of mechanistically different termination events in these studies. The heterogeneity of the ECs is amplified in the single-molecule observations.

Figure 2:

i) Prove that Rho is physically bound to the ISCs to show the existence of the "standby" mode. Non-specifically adsorbed Rho on the slides could be activated in the presence of NTPs and the longer RNA as it forms. What fraction of molecule showed termination after washing the slides? This would give an estimate of the wash-resistant fraction of Rho molecules. In the recent structural studies, Rho was shown to be bound to the EC having a long RNA (Said et al, 2021; Hao et al, 2020). Earlier Kalayani et al (2011) showed that Rho does not get associated with the ECs with short RNAs. The design of these experiments and the assumption that Rho binds to ISC is in direct conflict with earlier observations.

ii) What is the proof that Rho P279S will preoccupy the EC? This mutant is defective in ATPase or helicase activities but is capable of binding to single-stranded DNA or RNA in its PBS (Chaliisery et al., 2007). RNAP P279S-EC/RNAP binding assays have not been reported. Authors have to provide this binding data to design this assay.

iii) WT Rho can easily compete out P279S in these assays. So this assay will not distinguish between the two models.

iv) Please show representative fluorescence traces (like figure 1) as supplementary data to understand the plots.

Figure 3:

i) As per the single-molecule data, in most of the cases, "stand-by" models contribute less than 20%. Why do authors have to emphasize the existence of this model?

ii) How do data in figure 3, supplementary figures 4 and 5, prove that NusA/NusG strongly bind to ISCs? Please show direct binding assays.

Figure 4 and allied supplementary figures:

i) Descriptions of the experimental setup were very difficult to comprehend.

ii) It is important to show the termination pattern of these artificial templates in bulk solution assays. How did the authors determine the exact termination sites on these templates in single-molecule assays?

iii) In general, many artificial templates and non-characterized Rho-dependent terminators were used throughout. Their characterization as terminators is required in the regular in vitro transcription assays before using them in single-molecule studies.

Responses to the Comments of Reviewer #1:

Comments for the Authors

In this manuscript by Song et al., the authors use single-molecule fluorescence to observe extrinsic transcription termination by rho helicase. Their aim is to determine whether rho accompanies RNAP (stand-by termination) or loads onto RNA at a rut site (catch-up termination), and furthermore to determine whether these modes of termination occur via hypertranslocation of RNAP relative to DNA, or via shearing of RNA out of the RNAP. By attaching a Cy3 dye to the 5' end of the nascent RNA, and a Cy5 dye at the free DNA end, the authors can distinguish several events. They can observe RNAP begin to transcribe (by a decrease in Cy3-PIFE), and conduct either « recycling » termination (by a loss of Cy3 signal representing loss of RNA and a gain of Cy5-PIFE representing RNAP at the DNA end), « readthrough » (when Cy3 signal does not disappear and a Cy5-PIFE signal appears), and « decomposing » termination (by loss of Cy3 signal, which represents loss of RNA, with no simultaneous appearance of Cy5-PIFE). The relative fractions of these processes are then quantified and compared in situations where rho is pre-bound to RNAP (stand-by termination) or where rho has to move from the rut site to RNAP to remove it (catch-up termination). The main conclusion here is that both stand-by and catch-up termination coexist, and that catch-up termination leads to either decomposing or recycling termination (in a 5:1 ratio) whereas standby termination leads to decomposing but not recycling termination. Because recycling termination is enhanced when the AU content of the extrinsic terminator is increased, and because this change is expected to affect shearing but not hypertranslocation termination, the authors conclude that recycling termination occurs via shearing. Similarly the authors try to determine if decomposition occurs via hypertranslocation by measuring the effect of a 3-bp mismatch in the DNA:DNA hybrid, on the idea that a mismatch will inhibit duplex reformation and thus hypertranslocation but not shearing, however here the results do not conform to expectations and lead the authors to speculate on novel features of termination processes and RNAP structures.

Overall the authors convince this reviewer that both stand-by and catch-up activities of rho take place, and that when stand-by termination is blocked by prebinding a catalytically dead rho to RNAP, catch-up termination can still take place. Finally, RNAP recycling via rho-dependent catch-up termination appears to take place via shearing and catch-up termination can lead to decomposition or recycling but standby termination can lead only to decomposition. At the same time, the authors do not resolve the question of whether classical « decomposing » rho-dependent termination occurs via shearing or hypertranslocation – the looming question in the field as indicated at the start of the manuscript – and because the vast majority of rho-dependent termination events occur via decomposing and not recycling (itself a phenomenon which has only been described recently) this is somewhat unfortunate and reduces the impact of the work.

[Our response] We thank the Reviewer for thorough reading and favorable evaluation of our manuscript.

From a technical standpoint, this reviewer is also concerned about the reaction conditions used in this manuscript. In particular, the authors use an anti-bleaching solution based on the PCA/PCD system, which is known to rapidly acidify reaction conditions and lead to non-specific interactions (see Swoboda et al., ACSNano 6: 6364-6369 (2012)). Combined with only 10 mM of Tris buffering agent this seems potentially risky, and a pyranose oxidase antibleach seems like a better choice. How are the authors certain that this is not causing problems in their assay?

[Our response] Possible pH change during single-molecule fluorescence imaging is an important issue. According to the reference that the Reviewer mentioned, as shown in the figure below, the pH-8 buffer (10 mM Tris-HCl, pH 8.0, 10 mM MgCl₂, 150 mM KCl, 1 mM dithiothreitol, 5 mM PCA, 100 mM PCD, and saturated Trolox) does not suffer much acidification over time (blue dotted line), unlike pH-7 buffer with PCD (red dotted line) or any-pH buffer with GOC (dashed lines), which suffer much acidification over time. In fact, we used 40 mM Tris-HCl rather than 10 mM in all experiments, and it is corrected in the revised manuscript.

In order to make it sure, we additionally performed control experiments using POC, which shows even less acidification than PCD around pH 8, and we observe no significant difference in termination efficiency (TE) between the PCD and POC usages in the absence (a) or presence (b) of ρ , as shown below.

Minor points:

1. Related to the reaction conditions discussed above, it is surprising that the runoff times are so long (on the order of 200 seconds). Given the transcript is only ~200 bases, this would suggest the polymerase is only moving at approximately one base per second. This is at least a factor of ten slower than what is expected. Perhaps this is related to the pause at the terminator, but perhaps the reaction conditions are far from optimal? Have the authors carried out measurements for the runoff time in the absence of the terminator to show that their RNAP is working normally?

[Our response] It has been reported that *mgtA* terminator has an unusually long pause at the termination site [Hollands *et al.*, *PNAS* (2014) 111, E1999-2007]. To confirm that the long runoff time of *mgtA* terminator is mainly due to the long pause at the termination site, we performed additional experiments using an *mgtA* mutation (shown to the right) that greatly shortens the pausing time in the previous paper by Hollands *et al.*

[*PNAS* 2014].

As shown to the right, the runoff time of the mutant (36 s) is greatly reduced compared to that of the wild type (233 s).

By contrast, the runoff times of the other terminators are much shorter than that of *mgtA* terminator, as shown in Fig. 4d-e of our manuscript.

2. Is it possible that termination recycling is undersampled, for instance if the polymerase diffuses upstream (where there is no Cy5) rather than downstream of the terminator? How can the authors rule this out?

[Our response] Some post-terminational RNAPs diffusing in not only upstream but also downstream directions fall off DNA too soon before reaching the downstream end to cause Cy5 PIFE. These events are counted as decomposing rather than recycling, because termination events with Cy5 PIFE are counted as recycling termination and those without it as decomposing termination. Therefore, decomposing termination is practically defined as both simultaneous and near-simultaneous dissociations of RNA and RNAP, as described on lines 74-76 of page 3 and lines 135-136 of page 5, and recycling outcome refers only to long DNA retention of RNAP reaching the downstream end.

Post-terminational RNAPs reach the downstream end in an average of 3 seconds but can retain on DNA for an average of >34 seconds. The DNA retention time of fluorescently labeled RNAP was previously measured under varying salt concentrations as published in the Supplementary Figure S3 of our previous paper by Kang *et al.* [*Int. J. Mol. Sci* (2021) 11, 450], and can be approximated to be at least 34 s on the DNA templates under the ionic conditions used in this study. Therefore, roughly speaking, the post-terminational RNAP falling off DNA within ~3 s is regarded as decomposing, and longer retentions are regarded as recycling.

This is not much to affect the quantitative picture. Because post-terminational RNAPs fall off DNA too soon more frequently as the downstream end is farther away from the termination site (TS), how much of RNAP falls too soon can be estimated from a correlation of PIFE occurrence with the TS-end distance. This correlation was measured and published as Fig. 4d in our previous paper by Kang

et al. [*Nat. Commun.* (2020) 11, 450] and is shown to the right. The TS-end distance in our *mgtA* terminator template of this study is 46 bp, and the PIFE occurrence percentage can be approximated to be 91% according to the plot, so ~9% of RNAPs fall off DNA before causing Cy5 PIFE at the downstream end. This falls within the error range. For example, in the holistic single-molecule assay, relative frequency of recycling termination with *mgtA* terminator is $8.6 \pm 2.0\%$ as described on line 149 of page 5 in the revised manuscript, so 9% of 8.6 is 0.8 and much less than the error of 2.0.

3. How long does Cy5 PIFE last for in runoff – does the polymerase stay stuck indefinitely to the DNA end once it reaches it?

[Our response] As shown to the right, the recycling RNAPs reaching the downstream end dissociate with a finite DNA-binding lifetime of 267 s on average, as it is reflected in duration of PIFE exhibited by the Cy5 that is labeled at the downstream end, which is the runoff position.

4. It is a bit difficult to see the low concentration rho points in the termination efficiency graphs, and it is therefore a bit hard to see how standby and catch-up sum to the holistic assay.

[Our response] We redraw the Fig. 1e, Fig. 2b and Fig. 3b as semi-log plots with a base-10 log-scale on x-axis in the revised manuscript. We hope they are now improved enough.

5. The authors should better specify the number of events they are tallying and analyzing. They state that most data are from three experiments on over fifty molecules, but are multiple events measured on the same molecules or is there only one event from each molecule for instance?

[Our response] Only the first round of transcription is monitored. The second and subsequent rounds including reinitiations are not monitored because the Cy3-labeled ApU is washed away at the immobilized complex-washing step before resumption of transcription. This is newly described in the revised manuscript on lines 137-141 of page 5, starting a paragraph with "Reinitiation could occur by recycling RNAP after termination ...". The number of molecules used for data analysis are newly provided in Supplementary Tables 2-4 and Source data file.

Responses to the Comments of Reviewer #2:

Comments for the Authors

The authors present the results of their single molecule fluorescence-based assay looking at the pathway (or pathways as it turns out) of rho-dependent termination. Their data provide evidence that multiple distinct outcomes can be observed and may help unify previous reports supporting one of various models. These models principally RNA-centric or RNAP-centric recruitment of Rho and whether the polymerase dissociates or remains bound to the DNA template. The work nicely points out that molecular mechanisms don't necessarily only proceed on a linear pathway and also goes further to show that the fraction of individual outcomes can change depending on template sequence. The work is of interest to the field and provides a unique view into the molecular processes underlying termination in bacteria. I would recommend publication subject to my major points below:

[Our response] We thank the Reviewer for favorable evaluation of our manuscript.

Major points

(1) The aspect of pausing during termination is not adequately discussed or analyzed. However, the pause time is a crucial step which gives the time needed for termination to occur. The first measurement that should be made is a measurement of the pause time in the presence and absence of rho in both the "catch-up" and "stand-by" assays. This

can be done by comparing a DNA template lacking the termination sequence to the templates already described. Does the presence of Rho in stand-by mode change the pause kinetics? If not, then the pause part of the mechanism can be placed upstream of any Rho-dependent bifurcation of mechanism. This will allow the authors to dissect the times they measure for termination in the different situations on top of the sequence dependent termination pause.

[Our response] We agree with the Reviewer that the mechanistic understanding of pausing in transcription termination is an important issue, and we are pursuing this in a separate research project. We are afraid that this issue is too big and complicated to be included in the scope of this manuscript.

(2) Controls in the absence of a promoter sequence and/or in the absence of sigma factor need to be performed to show that the authors are observing truly promoter-dependent transcription complexes.

[Our response] In our additional experiments following this suggestion, we confirm that the elongation complex (EC) formation efficiency significantly decreases in the absence of the promoter or sigma factor, as shown to the right and below.

EC formation was measured as the ratio of the number of Cy3/Cy5 co-localized molecules to that of total molecules with Cy5 signal.

σ	Promoter	EC (%)	N in replicate experiments
no	no	7.5±1.6	5708 = 582 + 572 + 572 + 615 + 557 + 500 + 540 + 577 + 624 + 569
σ^{70}	no	5.2±1.9	3692 = 350 + 353 + 419 + 373 + 367 + 361 + 372 + 378 + 390 + 329
no	T7A1	27.0±1.4	3001 = 587 + 647 + 617 + 549 + 601
σ^{70}	T7A1	74.8±3.3	2939 = 589 + 597 + 585 + 586 + 582

(3) In Fig. 2B, a plot of the total termination efficiency along with the fractional plots would provide a clearer picture of what is going on.

[Our response] The figure that is revised following the suggestion is shown here to the right. Because the recycling termination is negligible at every concentration of ρ used, the total termination data mostly overlap with the decomposing termination data, making the figure quite cumbersome. If the Reviewer and the editor kindly allow, we would like to keep the original version of this figure but add a sentence "Because recycling is virtually null throughout a wide range of ρ concentrations, the decomposing data overlap with their sum data (not shown) in Fig. 2b." in the revised manuscript on lines 177-178 of page 6.

Additionally, the Fig. 2b legend is revised by addition of a sentence "Recycling is virtually null throughout a wide range of ρ concentrations, so the decomposing data overlap with their sum data (not shown)." on lines 602-604 of page 21. However, if the Reviewer and the editor do not agree, we will replace the figure.

(4) Are the times reported as to when termination occurs from when NTPs are flowed? Should these times be thought of as the sum of the termination pause + whatever other molecular transitions stimulated by Rho? Estimates or, even better, measurements of initiation time, elongation time, and pause times will provide context for these measurements. These times also seem quite long. Are they commensurate with other measurements of the kinetics of termination?

[Our response] As suggested, added to the revised manuscript is a sentence "Termination was timed as the interval from the Cy3 PIFE diminishing timepoint to the Cy3 signal vanishing timepoint, whereas readthrough was timed as the delay from Cy3 PIFE diminishing to Cy5 PIFE starting" on lines 161-164 of page 6.

Additionally, the Fig. 1f legend is revised by addition of two sentences "Termination was timed as the delay from Cy3 PIFE diminishing to Cy3 vanishing, whereas readthrough was timed as the delay from Cy3 PIFE

diminishing to Cy5 PIFE starting. These timings were estimated with the data fitting to single exponential functions." on lines 589-592 of page 20.

These times exclude initiation time but include elongation and pausing times, because we start to monitor fluorescence at 30 s before resumption of transcription. We think these times are commensurate with previous kinetic measurements. For example, the long runoff time with *mgtA* is due to unusually long pause [PNAS (2014) 111, E1999-2007], as we answer above to the minor point #1 by the reviewer #1.

(5) Extended data Fig. 5 shows that masking rho mutations suppress the full stand-by signal, but it is a small effect. How are the error bars calculated in this figure? They seem quite small! How many observations have been made under each condition? This comment applies to other figures as well. The number of observations need to be indicated.

[Our response] The masking by ρ^{P279S} mutant is complete unlike partial masking by ρ^{G51V} or ρ^{Y80C} mutant, and we use only ρ^{P279S} mutant in our catch-up assays. We now additionally explain this in the revised manuscript by adding a paragraph "For this purpose, three previously characterized ρ -mutants, ρ^{P279S} , ρ^{G51V} , and ρ^{Y80C} were confirmed to be completely inactive exhibiting only background termination (Fig. 1d), and tested for their masking activity in a modified stand-by assay where we pre-incubated ISCs with an inactive ρ mutant, washed away the unbound, added wild-type ρ , washed away the unbound again, and resumed transcription with NTPs alone (Supplementary Figure 7). When ρ^{P279S} is pre-bound, it is hardly replaced by subsequently added wild-type ρ , exhibiting full masking, whereas ρ^{G51V} and ρ^{Y80C} mutants exhibit only partial masking." on lines 192-198 of page 7.

Furthermore, the Extended data Fig. 5 (now Supplementary Fig. 7) is revised such that the ρ -dependent termination efficiencies (TEs) are estimated by subtraction of the ρ -independent background TE from the measured raw TEs. Additionally, a scheme of the modified stand-by assay is added as Supplementary Fig. 7a.

The background subtraction applies to all TEs throughout the revised manuscript. Additionally, the information on how the errors are estimated is shown throughout the revised manuscript and collectively in Supplementary Table 2.

(6) I want to commend the authors on performing their assay on multiple terminators and pointing out that outcomes change!!! This is great!

[Our response] We thank for the appreciation.

(7) I don't think one should use termination efficiency (normalized TE is probably even worse) as a measure of binding as the efficiency could change within the context of a bound complex.

[Our response] If the Reviewer refers to the supplementary Figure S4 (now Supplementary Figure 6), we agree that TE does not measure only ρ binding to RNAP. For example, some of RNAP-bound ρ molecules may be incompetent for termination, although the competency/incompetency ratio may or may not be changed by addition of NusA/G. The relevant sentences are now changed in the revised manuscript. The revised sentences are "As the time ..., indicating that activity lifetime of ρ -RNAP complex is finite but long enough to reveal stand-by termination." on lines 183-186 of page 6 and "Expectedly, the activity lifetime of stand-by ρ is prolonged with NusA/G than without them (Supplementary Fig. 6), ..." on lines 224-225 of page 8.

(8) What are the implications of the distinct timings reported? Presumably, these mechanisms will be engaged in a sort of kinetic competition with each other. In the cellular environment, does one expect there to always be a rho in stand-by mode based on concentrations?

[Our response] We think the stand-by termination occurs *in vivo*, because the fraction of stand-by termination out of total termination events, which is estimated by the ratio of stand-by termination to the sum of stand-by and catch-up terminations, is around ~40% and not

much sensitive to variation of ρ concentration as low as a sub-nanomolar concentration, as shown to the right. Regardless of ρ concentrations, a certain portion of it would be in stand-by mode in the cells as it binds RNAP even before *rut* site is synthesized to generate catch-up ρ . Then, their independent, competitive or collaborative actions would follow.

(9) With the stand-by rho positioned and extra added at NTP chase, shouldn't we expect stand-by to dominate? Why is it slower than "catch up"?

[Our response] This is indeed an interesting and important question. According to a conventional reasoning, stand-by termination would be speculatively faster and more efficient than catch-up termination, but our observation is in stark contrast to that speculation, as described in a new paragraph on lines 251-256 of page 9, in a new sentence in the Discussion section on lines 342-343 of page 11, and in Fig. 7 legend on lines 699-701 of page 27 in the revised manuscript.

At the moment, we only guess that stand-by ρ on RNAP is not ready or in active state by itself for binding *rut* site or releasing RNA. In fact, we are engaged in pursuing this issue that is too complicated and lengthy to be included in this manuscript.

Minor points

(1) To what degree are Cy3/Cy5 signals co-localized to assure that the Cy3 signal is DNA-dependent?

[Our response] We chose the individual complexes not only with Cy3/Cy5 co-localization but also with Cy3-PIFE change. Thus, the complexes that show Cy3 signal without dependence on DNA were excluded from our data analysis. Because Cy3-PIFE starts in the initially stalled complexes but diminishes as transcription resumes, Cy3-PIFE diminishing guarantees Cy3 signal's DNA-dependence. Roughly three quarters (73%) of Cy5 spots are colocalized with Cy3 signal.

(2) Caption in ED Fig. 6 says, "background efficiencies were measured with rho", but shouldn't this read, "without rho"?

[Our response] The error is now corrected in the revised manuscript. Thanks.

Responses to the Comments of Reviewer #3:

Comments for the Authors

The manuscript entitled, "Molecular ... termination", by Song et al., describes in vitro single-molecule measurements of Rho-dependent transcription termination process using an immobilized DNA template on a glass slide following the fluorescence signals emanating from the probes tagged at the DNA and the nascent RNA ends. The authors attempted to incorporate different heterogeneous behavior of the transcription machinery on the slide surface into different transcription termination models of Rho-dependent termination. I have many serious concerns about the experiments and their interpretations.

We thank the Reviewer for critically reading our manuscript.

General comments:

1. Many experiments have serious flaws in design and lack proper controls (see below under specific comments). To me, data were over-interpreted to fit into some mechanistic models.

We address all the concerns of the Reviewer as shown below.

2. Except for λ tR1 and trp t' , all the other terminators used are not properly characterized in vitro.

[Our response] All the terminators we used have been characterized in the following papers.

(1) Said, N. et al. Steps toward translocation-independent RNA polymerase inactivation by terminator ATPase ρ . *Science*. 371, eabd1673 (2021) (λ tR1 terminator).

(2) Epshtein, V., Dutta, D., Wade, J. & Nudler, E. An allosteric mechanism of Rho-dependent transcription

- termination. *Nature* 463, 245–249 (2010) (*trp-t'* terminator).
- (3) Hao, Z. et al. Pre-termination transcription complex: structure and function. *Mol. Cell* 81, 281–292 (2020) (*trp-t'* terminator).
 - (4) Hollands, K. et al. Riboswitch control of Rho-dependent transcription termination. *Proc. Natl. Acad. Sci. U. S. A.* 109, 5376–5381 (2012) (*mgtA*, and *ribB* terminators).
 - (5) Hollands, K., Sevostiyanova, A. & Groisman, E. A. Unusually long-lived pause required for regulation of a Rho-dependent transcription terminator. *Proc. Natl. Acad. Sci. U. S. A.* 111, E1999–E2007 (2014) (*mgtA* terminators).
 - (6) Bastet, L. et al. Translational control and Rho-dependent transcription termination are intimately linked in riboswitch regulation. *Nucleic Acids Res.* 45, 7474–7486 (2017) (*rho*, *ribB*, and *mgtA* terminators).
 - (7) Silva, I. J. et al. SraL sRNA interaction regulates the terminator by preventing premature transcription termination of rho mRNA. *Proc. Natl. Acad. Sci. U. S. A.* 116, 3042–3051 (2019) (*rho* terminators).
 - (8) Matsumoto, Y., Shigesada, K., Hirano, M. & Imai, M. Autogenous regulation of the gene for transcription termination factor rho in Escherichia coli: localization and function of its attenuators. *J. Bacteriol.* 166, 945–958 (1986) (*rho* terminators).
 - (9) Barik, S., Bhattacharya, P. & Das, A. Autogenous regulation of transcription termination factor Rho. *J. Mol. Biol.* 182, 495–508 (1985) (*rho* terminators).
 - (10) Brown, S., Albrechtsen, B., Pedersen, S. & Klemm, P. Localization and regulation of the structural gene for transcription-termination factor rho of Escherichia coli. *J. Mol. Biol.* 162, 283–298 (1982) (*rho* terminators).

Hence, regular in vitro transcription termination assays in bulk using radio-labeled NTPs is necessary to identify the termination sites, the efficiency of termination, termination kinetics, etc., to validate the observations and interpretations of the single-molecule experiments.

[Our response] Following the suggestion, we additionally performed in vitro bulk termination experiments for all ρ -dependent terminators used in our work without and with NusA/G. We add a sentence "Their ρ -dependent termination activities are confirmed in this study by using in vitro bulk termination assays (Supplementary Fig. 8), and consistent with previous reports^{7,9,12,18-19,26-30}." in the revised manuscript on lines 209-211 of page 7. Their data are shown in Supplementary Fig. 8.

3. The authors unnecessarily coined new terminologies like, catch-up, stand-by, holistic assays, etc. that complicated the descriptions. Terminologies like "RNA-dependent", "RNAP-dependent pathways", "kinetic coupling" are well established in the field and the literature. There is no need for new jargon.

[Our response] The assays we refer to as catch-up, stand-by, and holistic assays are all new single-molecule assays that have not been previously developed or described. All these are distinct from each other and mentioned so many times in the manuscript. We cannot think of better alternatives without coining new terms for them.

Additionally, our new terms 'catch-up' and 'stand-by' are related only with pre-terminational pathways rather than the entire processes associated with ρ -dependent termination, which additionally include terminational release of RNA and post-terminational outcomes. In order to minimize possible confusion, we now use catch-up/stand-by pre-terminational 'mechanisms' or 'modes' rather than such 'models' throughout the revised manuscript after their first appearance in the second and third paragraphs of the Introduction section. In other words, the catch-up mode refers to the pre-terminational step of the so-called kinetic-coupling, RNA-dependent, RNA-centric, and tracking models of ρ -dependent termination. Likewise, the stand-by mode refers to the pre-terminational step of the so-called RNAP-dependent, EC-centric, and allosteric models. Incidentally, usage of 'RNA-dependent' and 'RNAP-dependent' is not recommended because these names imply that ρ or ρ -dependent termination depends on only RNA or RNAP in each model, which is not the case, and because they are little distinguishable from each other with difference only in one-letter P.

4. The overall descriptions of different experiments are too sketchy and were very difficult to comprehend in many

places. A thorough re-writing is required.

[Our response] The originally submitted manuscript was written under very tight limitation of text length and word counts, and many descriptions had to be short and could be too sketchy for some readers. As the manuscript is now transferred to *Nature Communications* with much more relaxed limitation on text length, we choose to rewrite the entire manuscript. Literally every paragraph is revised, and descriptions are much more detailed throughout the manuscript.

The four previous figures are now split into seven figures, and all figure legends are much expanded with more details. Three figure panels (Figs. 5b, 5d, and 6c) move from the supplementary file to the main text, and a new figure panel (Fig. 4f) is added in the revised manuscript. The Fig. 7 is totally redrawn.

Specific comments:

Figure 1:

i) Show in vitro transcription data in solution with all the templates used in the study to identify the termination sites and to check whether they are matching with the fluorescence signals obtained in the single-molecule studies. Rho terminates over a wide range at multiple sites, which are needed to define for the uncharacterized terminators like, mgtA, rho, and rib. Heterogeneity observed in the single-molecule assays could be because of that. Speed of RNA or RNAP release could depend on the sequence and pausing states at each of these sites. Multiple termination events do not mean the existence of different mechanisms.

[Our response] As shown in our additionally-performed experiments (Supplementary Fig. 8), *mgtA* and *ribB* terminators exhibit a single focused major termination site, while *λtR1*, *trp-t'*, and *rho* terminators exhibit multiple dispersed termination sites, consistent with previous papers. All these terminators exhibit molecular heterogeneity, so it is unlikely that mechanism heterogeneity is associated with multiplicity or dispersion of termination sites.

We do not simply observe multiple termination events, but we do observe different mechanisms. For example, termination events by catch-up ρ are distinguished from those by stand-by ρ in our separate catch-up and stand-by assays. Simultaneous and near-simultaneous dissociations of RNA and RNAP are distinguished from temporally separate, sequential dissociations of them in fluorescence change patterns. Additionally, RNA shearing mechanisms are experimentally discerned from RNAP displacing. Regardless of whether termination occurs in single or multiple termination sites, experimentally distinguishable mechanisms are observed with all five different terminators that exhibit either single or multiple termination sites.

ii) A control transcription assay without Rho is needed.

[Our response] The negative control experiment is already included in Fig. 1d. In order to reduce possible overlook, we add a new phrase "in a negative control experiment" in front of "without ρ " in the revised sentence on lines 141-142 of page 5. In all the additionally-performed experiments, the negative controls are included as shown in Supplementary Fig. 8.

iii) No statistics have been provided describing the number of molecules that exist in each type of event.

[Our response] Following the suggestion, in the revised manuscript, we newly add Supplementary Tables 2-4 and Source data file describing the number of molecules used for every data analysis.

iv) Two events occur during the termination process, RNA release and RNAP release. There is no need to use terminology like "decomposition".

[Our response] The two dissociation events were not experimentally distinguishable until recently. Only last year, two independent single-molecule studies including ours developed experimental assays to measure the two events separately [references 15 to 17 of the revised manuscript]. While 'termination' refers to only RNA release rather than RNAP release, outcomes of termination could be three-fold. (1) When RNA release precedes RNAP dissociation, consequently RNAP 'recycles' on DNA for reinitiation as we and others reported

previously [references 15 to 17]. (2) Alternatively, RNA release synchronizes with RNAP dissociation. This outcome is called here as one-word 'decomposing' rather than any multiple-word terms or their unfamiliar abbreviations because it is mentioned so many times in the manuscript. (3) The third, intangible possibility that RNA release follows RNAP dissociation has not been observed in any intrinsic or ρ -dependent termination event. This scenario would not be possible because RNA-DNA hybrid is little stable without RNAP. Once RNAP is dissociated, instantly DNA would be rewound to release RNA.

In summary, the two observed outcomes of termination need to be distinctly described by concise and precise terms. We cannot think of better alternatives than 'recycling' and 'decomposing' as they correctly describe the outcomes, are the simplest with one word, and can be used as noun and adjective.

Slow-release of RNA coincides with the slow release of RNAP in 3rd type (1C) of event, which could also mean two events occur simultaneously. As there are multiple termination sites, why step-wise disappearance of Cy3-signal was not observed?

[Our response] In single-molecule experiments, simply it is not possible that Cy3-RNA is released or Cy3 disappears twice or more from a single transcription complex in a stepwise manner.

The transcription elongation process itself is heterogeneous, which is reflected in these assays.

[Our response] Elongation process is not directly observed in our setting of single-molecule assays.

v) The PIFE signal went down after some time, which means RNAP may not be recycled immediately. To prove the recycling of RNAP, the second round of transcription events with the same RNAP has to be shown experimentally.

[Our response] While recycling refers to retention of RNAP on template DNA after terminational release of RNA, reinitiation by recycling RNAP on the same DNA molecule has been demonstrated in three previous papers [references 15-17].

vi) From how far the PIFE could be observed? Can RNAP stalled/paused at the termination sites induce PIFE? These needed to be clarified before interpreting the data.

[Our response] According to Fig. 4 of the paper by Hwang et al. [*Chemical Society Reviews* (2014) 43, 1221-1229], as shown here to the right, PIFE-sensitive linear distance is 0 ~ 4 nm, which is about 12 bp or less. In the case of *mgtA* terminator, the termination site is 46 bp away from the downstream end, so RNAP at the termination site can hardly induce PIFE of Cy5 on the end. Additionally, in each of the other terminators, the DNA downstream end is sufficiently far away from the most downstream (last) termination site.

DNA template	The last termination site	DNA downstream end	Distance (bp)
mgtA	+218	+264	46
rho	+265	+340	75
rbiB	+279	+300	21
trp t'	+207	+236	29
λ TR1	+397	+418	21

vii) As there are multiple termination sites, depending on the release kinetics of RNA and RNAP at each of these sites, don't we expect to see these different types of events at different sites?

In general, ECs are highly heterogeneous under in vitro conditions, so heterogeneity at termination sites is expected. Some ECs are refractory to termination, some are slow releasing and some are fast releasing type. Hence, there is no need to introduce new terminologies like recycling, decomposition, holistic assays, etc., which confuses the reader. I am not at all convinced that these data support the existence of mechanistically different termination events in these studies. The heterogeneity of the ECs is amplified in the single-molecule observations.

[Our response] Determination or choice of termination sites would be much more complicated than the

Reviewer mentioned. It will depend on pausing, elongation and termination mechanisms including their kinetics not to mention their complex interactions.

One should not necessarily assume different types of events are expected at different sites. The same type of events can occur at different sites, simply depending on kinetics. Likewise, different types of events can occur at the same site depending on mechanisms.

We observe diverse mechanisms and outcomes of termination from individual transcription complexes using diverse assays. These different mechanisms, outcomes and assays need to be discerned by precise and concise terms. Incidentally, single-molecule experiments have been proven to be a powerful tool to correctly assess the mechanistic heterogeneity of various biochemical reactions.

Figure 2:

i) Prove that Rho is physically bound to the ISCs to show the existence of the “standby” mode. Non-specifically adsorbed Rho on the slides could be activated in the presence of NTPs and the longer RNA as it forms. What fraction of molecule showed termination after washing the slides? This would give an estimate of the wash-resistant fraction of Rho molecules. In the recent structural studies, Rho was shown to be bound to the EC having a long RNA (Said et al, 2021; Hao et al, 2020). Earlier Kalayani et al (2011) showed that Rho does not get associated with the ECs with short RNAs. The design of these experiments and the assumption that Rho binds to ISC is in direct conflict with earlier observations.

[Our response] In stand-by assays, the initially-stalled complexes (ISCs) are immobilized and pre-incubated with ρ , and then excess ρ is washed away before resumption of transcription. The background stand-by termination efficiency (TE) is measured without addition of ρ by counting Cy3 signal disappearance with no Cy5 change. To address the Reviewer's concern, we performed a control experiment for the background stand-by termination by adding ρ (100 nM) before rather than after ISC immobilization. If ρ is nonspecifically adsorbed and remains on the slides even after extensive washing and gets activated by NTPs or RNA to induce termination, the amount of background termination would increase. Simply, the background termination is not increased, as shown below. RNAP-unbound ρ is not activated in our stand-by assays.

DNA template	Condition	Raw TE (%)	n in replicated experiments
mgtA	Without ρ	6.3±0.7	413=103+117+193
mgtA	With ρ before immobilization	6.7±1.4	657=75+162+194+120+106

ii) What is the proof that Rho P279S will preoccupy the EC? This mutant is defective in ATPase or helicase activities but is capable of binding to single-stranded DNA or RNA in its PBS (Chaliisery et al., 2007). RNAP P279S-EC/RNAP binding assays have not been reported. Authors have to provide this binding data to design this assay.

[Our response] Our data in the masking experiments (Supplementary Fig. 7) already shows that ρ^{P279S} pre-binds RNAP. If it does not bind RNAP at the stand-by site, the subsequently added wild-type ρ would bind there to mediate termination, which is not the case at all. One can see that the fully inactive ρ^{P279S} (Fig. 1d) clearly preempts ρ binding to RNAP. In order to make it more comprehensible, we add a stepwise scheme of the masking assay in Supplementary Figure 7a of the revised manuscript. Furthermore, the stable binding of P279S on EC was previously reported [*Nature* (2010) 463.7278, 245-249].

iii) WT Rho can easily compete out P279S in these assays. So this assay will not distinguish between the two models.

[Our response] Our data in Supplementary Fig. 7 already demonstrates that the ρ^{P279S} is not replaced by subsequently added wild-type ρ . If the ρ is replaced, termination should have been increased even a bit. Not even a residual level of termination is observed, so one can see that its masking effect is complete. By contrast, the other two fully inactive mutants ρ^{G51V} and ρ^{Y80C} (Fig. 1d) exhibit only partial masking probably because some molecules are replaced (Supplementary Fig. 7).

iv) Please show representative fluorescence traces (like figure 1) as supplementary data to understand the plots.

[Our response] Following the suggestion, we now additionally show five representative fluorescence time traces in Supplementary Fig. 5 of the revised manuscript.

Figure 3:

i) As per the single-molecule data, in most of the cases, “stand-by” models contribute less than 20%. Why do authors have to emphasize the existence of this model?

[Our response] As the Reviewer mentioned, the stand-by termination pathway is not a dominant one, but it is certainly present in all four terminators, although negligible in λ *tR1* terminator. It is necessary to report as much as observed, especially because it has long been discussed as part of the so-called RNAP-dependent, EC-centric, and allosteric models. We think the stand-by portion is not insignificant to rule out its possible role in regulation of gene expression.

ii) How do data in figure 3, supplementary figures 4 and 5, prove that NusA/NusG strongly bind to ISCs? Please show direct binding assays.

[Our response] NusA/G has been previously reported to make a stable complex with RNAP, for example in recent papers by Hao et al. [*Mol. Cell* (2021) 81, 281-292] and by Said et al. [*Science* (2021) 371, eabd1673]. The same preparation made by one of us was used in the studies by Kang et al. [*Cell* (2018) 173, 1650–1662] and Kang et al. [*Nat. Commun.* (2020) 11, 450]. Our single-molecule assay data in Fig. 4 and Supplementary Figs. 6 and 7 and our bulk transcription assay data in Supplementary Fig. 8 show some effects of NusA/G on transcription termination efficiencies, clearly demonstrating that at least a significant portion of RNAPs is affected by association with NusA/G. Most importantly, our conclusions are not affected by addition of NusA/G, and do not appear to be related with how strongly NusA/G binds RNAP or ρ .

Figure 4 and allied supplementary figures:

i) Descriptions of the experimental setup were very difficult to comprehend.

[Our response] Now the explanations about RNA shearing tests and RNAP displacing tests are fully expanded on pages 9 and 10. We hope the Review finds it more comprehensible.

ii) It is important to show the termination pattern of these artificial templates in bulk solution assays. How did the authors determine the exact termination sites on these templates in single-molecule assays?

[Our response] Termination sites cannot be and were not measured in our setting of single-molecule assays. They were determined in our in vitro bulk transcription experiments (Supplementary Fig. 8) and the results are consistent with the previous papers as described in the revised manuscript on lines 209-211 of page 7.

iii) In general, many artificial templates and non-characterized Rho-dependent terminators were used throughout. Their characterization as terminators is required in the regular in vitro transcription assays before using them in single-molecule studies.

[Our response] All the five terminators used in this study have been well characterized in the ten previous papers that are listed above. Again, in vitro bulk transcription experiments were performed using all five terminator templates with or without NusA/G as shown in Supplementary Fig. 8, and the results are consistent with the previous papers.

Reviewers' Comments:

Reviewer #1:

Remarks to the Author:

The authors have satisfactorily addressed my comments and I believe the manuscript warrants publication in Nature Communications.

Reviewer #2:

Remarks to the Author:

The authors have responded in detail to most of the comments and concerns of the reviewers. As before, I believe the work should be published. I do believe the manuscript still needs some work prior to acceptance as detailed below.

(1) I agree with Reviewer 3 regarding issues of terminology. Specifically, the term, "recycling" is likely to mislead readers. It suggests either that some process results in the re-use of the same polymerase which is not what is observed here. In fact, polymerases are ALWAYS recycled via the pathway of dissociation, re-binding sigma, and association with a new promoter.

Here, the polymerase simply doesn't dissociate. While others have shown that these polymerases may actually initiate new transcripts (i.e., re-initiate), that process is not explicitly observed or studied here. But to be honest, even in that case, I wouldn't really want to call it a recycling.

More precise naming of the pathways observed should be used. Perhaps RNA-only termination (what the authors call recycling) could be distinguished from full-termination (what the authors call decomposing)?

(2) The authors note that there is a discrepancy between their results and those previously reported where mutant rho completely blocks termination by WT rho. A rationale, perhaps due to different experimental conditions or due to differences in the sensitivity of single-molecule vs. ensemble assays, for this discrepancy should be provided.

(3) The description and comparison of the timing of the different mechanisms is a bit confusing. I highly recommend editing for clarity. It seems like it may be the case that only if the fastest mechanism does not occur can the secondary and tertiary mechanisms occur. For example, only if rho does not catch up, then either the standby or read-through mechanisms can occur. Much of the work focuses on this point, but the current schematic in figure 7 does little to nothing to convey the relative timing aspect.

(4) In the concluding paragraph it is stated that, "In this study, we show that rho concurrently follows catch-up and stand-by pre-termination settings." However, I am not convinced that "concurrently" is the right word here. Perhaps something like, "...it is shown that, while catch-up and stand-by modes are operational modes of termination, they occur on different time-scales and tend to lead to different results (i.e., RNA-only termination vs. full-termination)"

(5) Do the data imply that at high concentrations of rho in solution, rho binds the rut site more readily from solution than from the stand-by site? If so, this should be stated.

(6) What exactly is normalized decomposing or recycling TE% reported in Fig 6/7? Is this percent of all molecules observed with an initial Cy3/RNA signal? What is the normalization. This needs further explanation.

(7) The interpretation of the shearing/translocation experiments should be handled with more caveats. The effects described from the mutations introduced are fine, but there may be other effects that would contribute to one or the other mechanism as well. For example, while there doesn't appear to be a correlation in Fig 6B, there are wild swings in TE with a greater magnitude than that exhibited in the correlative change in 6A. Conclusions based on these experiments should be softened. They may be consistent with or suggest a particular interpretation, but they

are not without caveats.

Reviewer #3:

Remarks to the Author:

The revised manuscript entitled, 'Molecular heterogeneity.....routes', by Song et al. is now much-improved and most of the comments have been responded adequately. However, I have still some conceptual problems with the manuscript that requires to be addressed properly before it could be accepted for publication.

1. Lines: 99-104: Please comment on the un-vanished spots after adding NTPs (figures S1a, b).

Are those arrested ECs?

2. Lines 119, Figure S3a: How do the authors distinguish between recycling RNAP and those that are simply stuck at the end. End-binding of RNAP is a well-known in vitro phenomenon on a linear DNA template.

3. Lines 137-141; As the assays do not measure the second round of transcription by the recycled RNAP, it is better to add that "signal could also arise from arrested or end-bound RNAP".

4. The section on stand-by model: Why TE is only <20% in these experiments? What fractions of ISCs are bound to Rho under this condition? Association of Rho to ISC (with short RNA) in these assays directly contradicts the earlier published biochemical data (JMB 2011, 413, 548-560), where Rho was shown to be associated only if the EC contains a longer RNA. Even in the two recently solved Cryo-EM structures, Rho-EC stable complexes were obtained in the presence of longer RNA. Please comment on this.

5. The section starting from line 206: Stand-by model involves Rho-RNAP interactions; why contribution from this mechanism will vary in different terminators? The RNA sequences are different in these different terminators and hence should not affect the stand-by mode of termination.

6. Lines 274-282: If Rho imparts RNA-shearing, why does the mutant sequence with higher AU content show lower termination efficiency?

7. I think using a road-block (using EcoR1 or lac repressor) downstream of the termination sites would be a better experimental strategy to prove the existence of hyper-translocation mechanisms.

8. A general comment. The basic difference between the stand-by and the catch-up models is at the Rho-recruitment steps. Once Rho is recruited to EC, both the model envisages Rho-translocation along with the RNA and dissociation of the EC via RNA shearing or hyper-translocation mechanisms or by both. I did not understand why these two models would yield different extents of recycling and decomposition of RNAP.

9. "Disintegration" would be a better phrase instead of "Decomposition".

Response to the Comments of Reviewer #1

Remarks to the Author:

The authors have satisfactorily addressed my comments and I believe the manuscript warrants publication in Nature Communications.

[Our response] We thank the Reviewer for satisfaction.

Response to the Comments of Reviewer #2:

Remarks to the Author:

The authors have responded in detail to most of the comments and concerns of the reviewers. As before, I believe the work should be published. I do believe the manuscript still needs some work prior to acceptance as detailed below.

[Our response] We thank the Reviewer for favorable evaluation and address all the Reviewer's comments below.

(1) I agree with Reviewer 3 regarding issues of terminology. Specifically, the term, "recycling" is likely to mislead readers. It suggests either that some process results in the re-use of the same polymerase which is not what is observed here. In fact, polymerases are ALWAYS recycled via the pathway of dissociation, re-binding sigma, and association with a new promoter. Here, the polymerase simply doesn't dissociate. While others have shown that these polymerases may actually initiate new transcripts (i.e., re-initiate), that process is not explicitly observed or studied here. But to be honest, even in that case, I wouldn't really want to call it a recycling. More precise naming of the pathways observed should be used. Perhaps RNA-only termination (what the authors call recycling) could be distinguished from full-termination (what the authors call decomposing)?

[Our response] RNAP recycling on DNA is now specified as 'one-dimensional (1D) recycling.' Likewise, transcription reinitiation by such recycling RNAP on the same DNA molecule is now specified as '1D reinitiation.' These terms appear first on lines 73-75 of page 3 in this revised manuscript, and the changes in wording are marked in red. It is revised as "This post-terminational one-dimensional (1D) diffusion of RNAP is named as 1D recycling (shortened here as recycling), and the reinitiation by such recycling RNAP is named as 1D reinitiation."

Transcriptional termination should refer to the release of RNA product rather than RNAP enzyme from DNA template. This definition is evident in that historically transcription termination has been experimentally observed by release of radioactively labeled RNA, not RNAP, and that the termination site has been defined as the DNA position where RNA extension terminally stops regardless whether, where and when RNAP falls off DNA. In this regard, we disfavor the suggested terms, RNA-only termination and full termination, because they erroneously imply that termination requires dissociation of both RNA and RNAP from DNA.

It is not simple that RNAP remains on template after transcript alone is released, because it brings about another functional stage of transcription cycle after the termination stage. The post-terminational stage of RNA-free RNAP on DNA has been named as 'recycling stage' in two recent papers of ours [*Nat. Commun.* (2020) 11, 450; *Int. J. Mol. Sci.* (2021) 22, 2398] in analogy with the textbook term of translation, 'ribosome recycling stage,' during which ribosome remains and diffuses on mRNA after terminational release of nascent peptide product and facilitates translational 'reinitiation' on downstream open reading frames, providing a regulation mechanism.

The recycling stage of transcription is functionally relevant with the transcriptional reinitiation on the same template DNA molecule through 1D diffusion rather than three-dimensional (3D) dissociation/reassociation. This 1D reinitiation was previously observed and characterized in three recent studies including ours [*Nat. Commun.* (2020) 11, 448; *Nat. Commun.* (2020) 11, 450; *Int. J. Mol. Sci.* (2021) 22, 2398]. For example, the post-terminational 1D diffusion was characterized to involve hopping and sliding in both forward and backward directions with occasional flipping of RNAP on DNA. Furthermore, the 1D reinitiation has been experimentally verified to efficiently occur on the original promoter and nearby promoters that are located at upstream or downstream in the sense or antisense orientation.

Although the two terms, recycling and reinitiation, were adopted in specific contexts to refer to only the recycling on DNA and only the reinitiation by such recycling RNAP, respectively in the two previous papers of ours without any objection from reviewers and editors, we understand that these terms in general can each describe various other mechanisms such as 3D dissociation/association, partial disassembly of transcription complex, and retention of some initiation factors, among many others [for example, *Trends Biochem.* (2003) 28, 202; *Nat. Commun.* (2020) 11, 6418; *J. Biol. Chem.* (2021) 297, 101404]. Therefore, we now like to call such recycling and reinitiation by post-terminational RNAP diffusing on DNA specifically as '1D recycling' and '1D reinitiation,' respectively in order to make precise distinction from the other possible mechanisms of recycling and reinitiation.

2) The authors note that there is a discrepancy between their results and those previously reported where mutant rho completely blocks termination by WT rho. A rationale, perhaps due to different experimental conditions or due to differences in the sensitivity of single-molecule vs. ensemble assays, for this discrepancy should be provided.

[Our response] While we do not know the exact cause(s) of the discrepancy for sure, we can describe a difference of the assays in the revision. We now add two phrases into the sentence on lines 200-203 of page 7. "While the catch-up assay's experimental setup with pre-bound ρ^{P297S} is validated, both decomposing and recycling terminations are observed with catch-up ρ in our single-molecule assay (Fig. 3b, Supplementary Fig. 5b) in contrast to a previous report that pre-binding of the mutant ρ completely blocks the termination by subsequent wild-type ρ in a bulk assay⁹."

(3) The description and comparison of the timing of the different mechanisms is a bit confusing. I highly recommend editing for clarity. It seems like it may be the case that only if the fastest mechanism does not occur can the secondary and tertiary mechanisms occur. For example, only if rho does not catch up, then either the standby or read-through mechanisms can occur. Much of the work focuses on this point, but the current schematic in figure 7 does little to nothing to convey the relative timing aspect.

[Our response] Fig. 7 is now supplemented with a vertical time line colored in gray. The readthrough event is removed from the figure because its timing varies. The figure layout is additionally improved.

The relevant paragraph in the Discussion section on lines 342-347 of pages 11 and 12 is revised and now ends with "Lastly, stand-by ρ mediates the decomposing termination separately from catch-up ρ . Interestingly, stand-by ρ could bind RNAP earlier than catch-up ρ but does not achieve the same termination more readily."

Furthermore, the Fig. 7 legend is revised on lines 693-706 of page 28. For example, the sentence on lines 698-701 of page 28 is revised as "(2) If the first termination fails, the second termination can be taken by catch-up ρ . Transcription complex decomposes rapidly at once as both RNA and RNAP depart DNA upon termination. This second-route termination by catch-up ρ for decomposing always occurs later than ..."

(4) In the concluding paragraph it is stated that, "In this study, we show that rho concurrently follows catch-up and stand-by pre-terminational settings." However, I am not convinced that "concurrently" is the right word here. Perhaps something like, "...it is shown that, while catch-up and stand-by modes are operational modes of termination, they occur on different time-scales and tend to lead to different results (i.e., RNA-only termination vs. full-termination)."

[Our response] The suggested phrase is added to the concluding paragraph of the Discussion section on lines 374-375 of page 12 as "These routes apparently operate on different time scales and tend to lead to distinct outcomes."

We used the word 'concurrent' to mean running parallel but people can misunderstand it to mean occurring at the same time. We remove it throughout the revised manuscript and even the title to avoid possible misunderstanding. The revised title is "Molecular heterogeneity of Rho-dependent transcription termination proceeding in three routes" on lines 3-4 of page 1 and keeps the word limit (≤ 15 words) with 10 words.

Additionally, the word 'concurrent' is replaced by 'kinetically different' in the first sentence of the Fig. 7 legend on lines 691-693 of page 28. "Three kinetically different routes of ..."

Furthermore, the Abstract is accordingly revised as "... These mechanisms are stepwisely coupled ... and kinetically different. (1) The catch-up mode leads first to RNA shearing for recycling, and (2) later to RNAP displacing for decomposing. (3) The last termination is usually taken by the stand-by mode with displacing for decomposing. ..." on lines 35-39 of page 2, keeping the word limit (≈ 150 words) with 150 words.

(5) Do the data imply that at high concentrations of rho in solution, rho binds the rut site more readily from solution than from the stand-by site? If so, this should be stated.

[Our response] We have not directly measured the binding of ρ to RNA *rut* site. However, ρ up to 200 nM does not necessarily appear to bind *rut* more readily from solution than from the stand-by site according to our data that were already shown to the Reviewer #2 regarding the comment #8 in our first revision. The same data of ρ concentration variation in the x-axis are redrawn here to the right with the fraction of the catch-up termination (by Rho from solution) at the y-axis this time instead of the stand-by termination last time. The catch-up fraction out of the total terminations does not appear to increase with rising concentration of ρ up to 200 nM.

(6) What exactly is normalized decomposing or recycling TE% reported in Fig 5/6? Is this percent of all molecules observed with an initial Cy3/RNA signal? What is the normalization? This needs further explanation.

[Our response] As each ρ -dependent termination efficiency (ρ TE) is estimated by subtraction of the background TE

measured without ρ (TE_0) from the raw TE measured with ρ , the maximum of ρTE is no longer 1 but is $1-TE_0$, so ρTE is normalized back to the maximum of 1 by $\rho TE/(1-TE_0)$. For example, when TE_0 is 6% or 0.06, ρTE is divided by 0.94 to yield a normalized value. Three relevant sentences are revised as "... ρ -dependent TEs (= raw TE - background TE) are normalized to the maximum of 100% by dividing them with (1 - background TE) ..." in the Result section on lines 278-280 of page 9, in the Fig. 5 legend on lines 661-663 of page 26, and in the Fig. 6 legend on lines 676-679 of page 27.

(7) The interpretation of the shearing/translocation experiments should be handled with more caveats. The effects described from the mutations introduced are fine, but there may be other effects that would contribute to one or the other mechanism as well. For example, while there doesn't appear to be a correlation in Fig 6B (5D?), there are wild swings in TE with a greater magnitude than that exhibited in the correlative change in 6A (5C?). Conclusions based on these experiments should be softened. They may be consistent with or suggest a particular interpretation, but they are not without caveats.

[Our response] We agree with the Reviewer, and soften the sentence with 'more likely' on line 285 of page 10. The next sentence on lines 286-287 of page 10 is additionally revised as "The large uncorrelated variations in Fig. 5d alludes some mechanism(s) other than RNA shearing operate for decomposing termination, so the hyper-translocation ..."

Response to the Comments of Reviewer #3:

Remarks to the Author:

The revised manuscript entitled, "Molecular heterogeneity ... routes", by Song et al. is now much-improved and most of the comments have been responded adequately. However, I have still some conceptual problems with the manuscript that requires to be addressed properly before it could be accepted for publication.

[Our response] We thank the Reviewer for favorable evaluation and address all the Reviewer's comments below.

1. Lines: 99-104: Please comment on the un-vanished spots after adding NTPs (figures S1a, b). Are those arrested ECs?

[Our response] We add a sentence in the Supplementary Figure S1 legend. "In a, the molecules indicated by red circles could be transcriptionally inactive complexes or active complexes where termination or run-off has not occurred yet."

2. Lines 119, Figure S3a: How do the authors distinguish between recycling RNAP and those that are simply stuck at the end. End-binding of RNAP is a well-known in vitro phenomenon on a linear DNA template.

[Our response] In single-molecule assays, once RNAP falls off DNA, it diffuses away from DNA and seldom binds the DNA ends, unlike in bulk assays. Initially-free RNAPs are all washed away before elongation is resumed.

A phrase 'to diffuse away' is added to the sentence on lines 136-137 of page 5 as "Alternatively, RNAP only briefly remains on DNA upon termination and falls off DNA to diffuse away before reaching the end."

3. Lines 137-141; As the assays do not measure the second round of transcription by the recycled RNAP, it is better to add that "signal could also arise from arrested or end-bound RNAP".

[Our response] Following the suggestion, we revised the sentence on lines 129-131 of page 5 as "Only RNA is released at the termination site, ... to reach the downstream end and stick there for a while, causing Cy5 PIFE."

Any complex arrested during elongation does not release Cy3-RNA or exhibit Cy5 PIFE, so is not included in our analysis. We found very few such arrested complexes in our experimental settings.

4. The section on stand-by model: Why TE is only <20% in these experiments? What fractions of ISCs are bound to Rho under this condition? Association of Rho to ISC (with short RNA) in these assays directly contradicts the earlier published biochemical data (JMB 2011, 413, 548-560), where Rho was shown to be associated only if the EC contains a longer RNA. Even in the two recently solved Cryo-EM structures, Rho-EC stable complexes were obtained in the presence of longer RNA. Please comment on this.

[Our response] In contrast to the 2011 paper mentioned, a recent 2021 review paper by Hao, Svetlov and Nudler states that the recruitment of ρ to RNAP does not depend on the presence of RNA [Hao et al. (2021) *Transcription* 12, 171], citing their study on cryo-EM structure of pre-termination complex [Hao et al. (2021) *Mol. Cell* 81, 281].

5. The section starting from line 206: Stand-by model involves Rho-RNAP interactions; why contribution from this mechanism will vary in different terminators? The RNA sequences are different in these different terminators and hence should not affect the stand-by mode of termination.

[Our response] Our data with the five different terminators are consistent with that the stand-by ρ specifically interacts

with RNA *rut* site in addition to RNAP stand-by site. Besides, as far as we know, the so-called RNAP-dependent, EC-centric, and allosteric models starting with the stand-by mode do not exclude specific interaction between ρ and *rut*.

6. Lines 274-282: If Rho imparts RNA-shearing, why does the mutant sequence with higher AU content show lower termination efficiency?

[Our response] If the Reviewer refers to the decrease in decomposing termination with the AU100 mutant in Fig. 5b, in fact, its termination efficiency fluctuates greatly on sequence variation of the RNA-DNA hybrid at the termination site as shown in Fig. 5d. This large fluctuation alludes the presence of other mechanism(s), so a phrase is added on lines 286-287 of page 10. "The large uncorrelated variations in Fig. 5d alludes some mechanism(s) other than RNA shearing operate for decomposing termination, so the hyper-translocation model was tested ..."

We do not have an answer to the question yet but are currently trying to understand how the hybrid sequence variations affect decomposing termination in a follow-up study.

7. I think using a road-block (using EcoR1 or lac repressor) downstream of the termination sites would be a better experimental strategy to prove the existence of hyper-translocation mechanisms.

[Our response] This suggestion is more than welcomed, but we would like to try the suggested roadblock strategy in our future works. We do not think we need to prove the presence of hyper-translocation to support the major conclusions of our current study. Characterization of RNA release mechanism is much less necessary than determination of terminational outcomes in this study.

8. A general comment. The basic difference between the stand-by and the catch-up models is at the Rho-recruitment steps. Once Rho is recruited to EC, both the model envisages Rho-translocation along with the RNA and dissociation of the EC via RNA shearing or hyper-translocation mechanisms or by both. I did not understand why these two models would yield different extents of recycling and decomposition of RNAP.

[Our response] In response to this important comment, we add a new paragraph in the Discussion section on lines 348-353 of page 12. "The catch-up and stand-by pre-terminational modes would follow separate paths on different time scales and could build disparate structures of termination-proficient complex favoring distinct outcomes such as recycling on DNA and decomposing of complex. We speculate that even for the same decomposing termination, the catch-up and stand-by ρ 's could induce dissimilar changes in termination complex structure, because their termination timings often differ from each other (Fig. 4f)."

9. "Disintegration" would be a better phrase instead of "Decomposition".

[Our response] Distinction between the two words can be summarized as that disintegration is breaking up into parts and decomposition is breaking down into components according to many English dictionaries. RNA, RNAP and DNA are essential components of transcription complex rather than subcomplex parts or chemically broken parts. Accordingly, we tend to think that decomposition is more appropriate than disintegration for referring to that a complex is disassembled rapidly at once into its components.

Reviewers' Comments:

Reviewer #2:

Remarks to the Author:

I would thank the authors for their thoughtful responses to my concerns. I am now able to recommend publication of their interesting findings!

Reviewer #3:

Remarks to the Author:

The authors' responses to most of the comments are satisfactory and the concerns raised were addressed properly.

However, I have a strong reservation about the claims of non-requirements of RNA during the Rho recruitment process in the review of Hao et al. Both the recent Rho-EC structures contain ~100 nt long RNA, and there is no strong experimental evidence for Rho-EC complex formation in solution in the absence of nascent RNA.

I prefer authors to tone down their descriptions in this regard.

This manuscript may be accepted for publication now.

Response to the Comments of Reviewer #2

Remarks to the Author:

I would thank the authors for their thoughtful responses to my concerns. I am now able to recommend publication of their interesting findings!

[Our response] We thank the Reviewer for the recommendation.

Response to the Comments of Reviewer #3:

Remarks to the Author:

The authors' responses to most of the comments are satisfactory and the concerns raised were addressed properly. However, I have a strong reservation about the claims of non-requirements of RNA during the Rho recruitment process in the review of Hao et al. Both the recent Rho-EC structures contain ~100 nt long RNA, and there is no strong experimental evidence for Rho-EC complex formation in solution in the absence of nascent RNA. I prefer authors to tone down their descriptions in this regard. This manuscript may be accepted for publication now.

[Our response] The pertinent question is what fraction of the initially-stalled complexes (ISCs) are bound to Rho under the conditions of our stand-by termination assay where a relatively short, five-nucleotide long RNA is attached to the ISCs, which was put in this reviewer's previous comment #4 for the second revision. While Rho appears to bind at least some of ISCs to mediate the stand-by termination in the assay with 100 nM of input Rho, we have not measured what fraction of ISCs are actually bound, although the stand-by termination efficiency appears to approach a maximum plateau around 100 nM Rho in the titration experiment of Figure 2b. Therefore, the relevant description is toned down as suggested. The third sentence of the section entitled "Co-present catch-up and stand-by pre-terminational mode" is now revised as "Although it is not known what fraction of ISCs are bound to ρ under the conditions of this stand-by single-molecule assay, only pre-bound ρ in the stand-by mode can mediate termination because unbound ρ for catch-up mode is washed away." on lines 182-184 of page 6. The changes are marked in red.

In our response to this reviewer's previous comment #5, we did not make any change in the manuscript. Now we would like to revise a sentence and add a sentence on lines 224-227 of page 8. "The ρ -dependent TEs of the five terminators measured by the single-molecule holistic assays range from 12 to 51% as both catch-up and stand-by TEs vary on terminators (Fig. 4a, Supplementary Fig. 9). These results are consistent with that ρ in either mode interacts specifically with RNA *rut* sequence as well as with RNAP."

Furthermore, in our response to this reviewer's previous comment #2 'How do the authors distinguish between recycling RNAP and those that are simply stuck at the end,' we have already revised the sentence on lines 142-144 of page 5 by adding a phrase 'to diffuse away.' In order to make it more understandable in this third revision, we would like to additionally add a sentence "On the other hand, once RNAP falls off DNA, it readily diffuses away and seldom rebinds DNA even at the ends in this single-molecule assay." on lines 124-125 of page 5.